# Dubbing for Everyone: Cost and Data-Efficient Visual Dubbing using Neural Rendering Priors

## Abstract

Visual dubbing is the process of generating lip motions of an actor in a video to synchronize with given audio. Visual dubbing allows video-based media to reach global audiences. Recent advances have made progress towards realizing this goal. However, **existing person-specific models see only one frame of the actor and, therefore, lack the ability to capture identity in the form of characteristic motion and related idiosyncracies**, or they are expensive methods requiring off-putting **large data requirements** and costly model training. Our key insight is to train a large, multi-person prior network, which can then be adapted to new users. This method allows for **high-quality visual dubbing with just a few seconds of data**, that enables video dubbing for any actor - from A-list celebrities to background actors. We show that we achieve state-of-the-art in terms of **visual quality** and **recognizability** both quantitatively and qualitatively through two user studies. Our prior learning and adaptation method **is able to adapt to small datasets better than baselines**. Our experiments on real-world, limited data scenarios find that our model is preferred over **baseline models**.

## 1 Introduction

Dubbing is the process of translating video content from one language to another. For the most part, dubbing is performed only on the audio tracks, leaving the video unchanged. This results in a poor visual experience. Visual dubbing involves reconstructing the lip and mouth movements of the actor in a video to match new audio in a different language. When done correctly, visual dubbing transforms how global audiences watch video content filmed in non-native languages, allowing content creators to reach more viewers worldwide **(Fla)**.

We argue that any successful video dubbing method must be **high-quality**, **generalizable**, **recognisable** and **low-cost**. It must be **high-quality** so consumers are not distracted by the synthesized lips, avoiding the 'uncanny valley' effect. This requires good video quality and good lip sync. It must be **generalizable** in that all actors, from A-list stars to background actors **who only appear for a few seconds of high-quality dialogue**, should be dubbed effectively with as little as a few seconds of dialogue. An actor's idiosyncratic style should be **recognisable** in the dubbed video. For instance, the actor's lips and teeth should look the same in the dubbed video as in the real one. To be viable, a visual dubbing solution should also be low-cost.

Previous models are split between their focus on these criteria. Some produce excellent videos for a single actor under controlled conditions (e.g. Thies et al. (2020); Ye et al. (2023); Tang et al. (2022)). Such methods are high-quality and recognisable but will work only on the actor they are trained on. They also require significant training data, often at least 2 minutes. This makes them unsuitable for practical dubbing in scenes where actors may only be present for a few seconds. Other methods produce a low-quality but generalisable video (e.g. Wang et al. (2023a); Gupta et al. (2023); Guan et al. (2023). These methods can be applied to any audio and video cheaply, but the outputs are rarely of good visual quality and do not capture the style of the actors. For instance, they all produce overly generic teeth and mouth interiors; see Figure 5.

We propose Dubbing for Everyone. We create a model that meets ***all*** our criteria, allowing the high-quality, low-cost dubbing of all actors, including those with short roles. **The critical insight of our work**

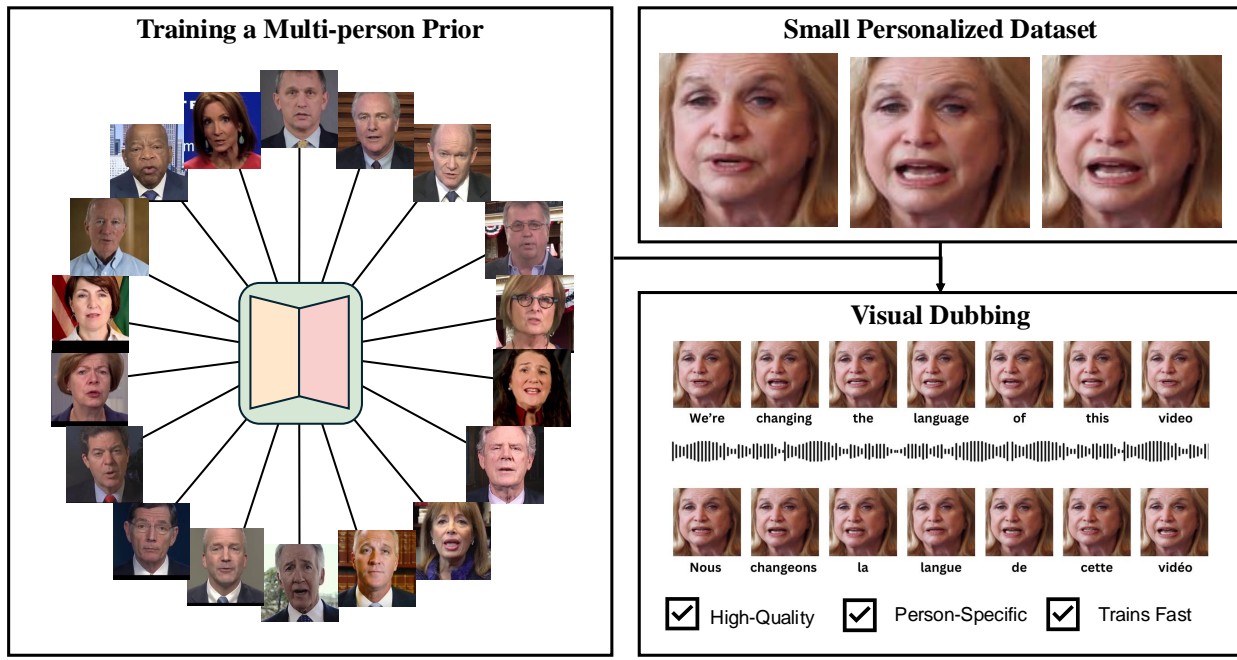

Figure 1: Our method, Dubbing for Everyone, allows for the reconstruction of lip movements when dubbing video from one language to another, using only a few seconds of training data. We do this by training a large-scale prior over many people, dramatically reducing the data requirements.

**is that Neural Rendering models can be de-coupled into person-generic and person-specific components, and that they benefit from a large-scale pre-training of generic ones**. We call this a prior network. Our prior network is trained across multiple actors and can generalise across identities. We also maintain actor-specific components, designed to guide the prior, that allow our model to adapt to new individuals and capture actor-specific nuance. In particular, we adopt a multi-stage approach based on neural textures (Thies et al., 2020; 2019).

Our model is **generalisable** due to the person-generic prior network training on a large dataset. We demonstrate this in Table 1 by using as few as 4 seconds of actor-specific data and further discuss this quality in Section 5.6. It is **high-quality** due to the person-specific adaptation making effective use of all data, we achieve state-of-the-art in this respect as is shown in Table 1. We find an order-of-magnitude speedup compared to **a baseline model trained without priors** (Section 5.4), leading to a similar order-of-magnitude cost reduction. By maintaining actor-specific components, or model is **more able to capture recognisable idiosyncrasies, as shown in Table 1** and does not appear overly generic. We validate this through a user study in Table 2. The novel contributions in this paper may be summarised as:

- We present **Dubbing for Everyone**, a visual dubbing model using person-generic and person-specific components, capable of producing **high-quality** and **idiosyncratic** results from just a few seconds of data.

- We train a prior deferred neural rendering network across many identities and learn actor-specific neural textures, allowing us to adapt our model to new identities. The training of the prior network allows for **data-efficent dubbing**, resulting in a significant reduction in data requirements compared to existing person-specific models.

- We propose a novel post-processing algorithm to remove artefacts in the border around the generated video. This improves perceived quality (Table 4).

- We perform an extensive evaluation to show that our method achieves state-of-the-art for **quality**(Table 1, Table 2, & Supplementary Video) and **recognisability**(Table 1), as well as being **an**

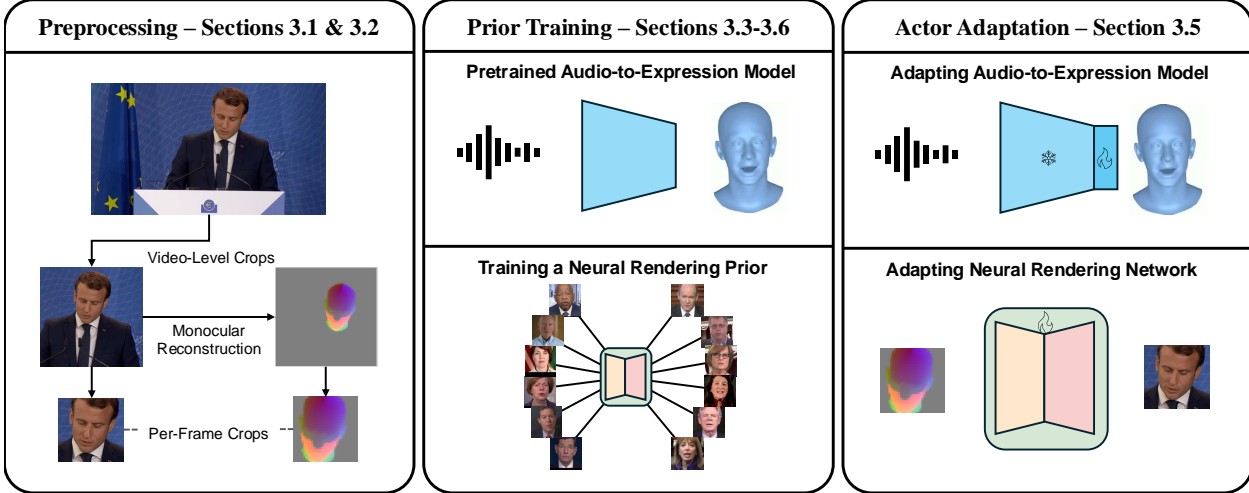

Figure 2: The pipeline of our method. We first apply preprocessing to our dataset (Section 3.2) to obtain 3D reconstructions, tightly and stably cropped to the face. We next obtain person-generic audio-to-expression and neural rendering models using multiple subjects (Section 3.4). Given a new subject, we then finetune both models for the given subject (Section 3.5).

> **order of magnitude cheaper** **than existing person-specific models** (Section 5.4 & Table 3) and **robust in few-shot scenarios.** (Section 5.6 & Table 1)

## 2 Related Work

Early works in visual dubbing (e.g. Bregler et al. (1997); Ezzat et al. (2002)) use various methods. However, for this work, we consider post-deep learning models. Two separate classes of visual dubbing exist person-generic and person-specific models.

### 2.1 Person Generic Models

Person-generic models differ from person-specific models **as they do not require any data of a given subject for training or fine-tuning. They only see a single reference frame given as input.** These methods are typically 2D-based. One class of these models uses some form of expert discriminator to achieve lip sync. Early methods (Prajwal et al., 2020; K R et al., 2019) use an encoder-decoder model to predict frames from audio and use random reference frames of the same person. These methods use adversarial training combined with an expert Syncnet (Chung and Zisserman, 2016), which predicts if video and audio are in or out of sync. Most person-generic models build upon this framework but replace some of these components. Some seek to replace the encoder-decoder model with transformers (Wang et al., 2023b) or vector-quantised models (Gupta et al., 2023). Others change the type of expert discriminator (Wang et al., 2023a; Sun et al., 2022) or replace the adversarial loss with a diffusion process (Shen et al., 2023; Stypulkowski et al., 2024; Liu et al., 2024; Zhang et al., 2024).

In either case, these models use only a single reference frame to encode the identity. This is a significant issue as a single image cannot contain enough information about appearance or talking style. For example, if the mouth is closed in the reference frame, it is impossible to predict what the interior should look like. Our work, in contrast, can use all available frames of the target person for fine-tuning, enabling us to capture idiosyncratic qualities.

## 2.2 Person Specific Models

Person-specific models are trained per person, usually under controlled conditions. As a result of this, they are typically much higher quality than person-generic models but cannot produce results for anyone other than the person they were trained on. It is very common for person-specific models to use some form of 3D supervision in order to improve the results. By introducing 3D priors, certain characteristics, such as the face shape or pose, can be controlled for.

One line of work builds upon the 3D Morphable Model (Egger et al., 2020; Blanz and Vetter, 2023). The 3DMM allows pose, lighting, shape and texture to remain constant, only changing the expression of the face. Some works achieve dubbing by having one actor provide the lip motions for another (Kim et al., 2018; 2019), while others generate the lip motions from audio (Thies et al., 2020; Saunders and Namboodiri, 2023; Wen et al., 2020; Song et al., 2022). **While some of these models train a prior for the lip motion, none do so for the appearance.**

Person-specific models share some essential qualities. They all produce high-quality output but come with significant data requirements **of around 2-5 minutes (Thies et al., 2020)**. In contrast, our method achieves similar quality using as little as 4 seconds of training data, thanks to our person-generic prior network training and person-specific adaptation.

The most similar work to ours could be considered to be StyleSync (Guan et al., 2023). This method performs visual dubbing and has demonstrated an ability to adapt to new identities using fine-tuning. **There are two models presented in StyleSync: StyleSync-G, which is person-generic and does not require any subject-specific data, and StyleSync-P, which uses a small amount of data for finetuning.** However, the model does not decouple person-specific and person-generic components, while ours does. This makes it less capable of capturing person-specific nuances (Table 1, Table 2).

## 3 Method

Our method builds upon the deferred neural rendering approach of Thies et al. (2019). The key to our method is the training of a prior deferred neural rendering network (Section 3.4), which is person-generic, and the adaptation to new actors using neural textures (Section 3.5). This method (Figure 2) requires an order of magnitude less data than previous neural textures approaches. Before training the prior network, we run a preprocessing stage, which involves cropping the video frames to the face region and performing monocular reconstruction (Section 3.1) to get a parameterized 3D representation of the face.

Using an existing speech-driven animation model (Thambiraja et al., 2023), we are able to control the 3D model and, in turn, alter the lip motions of a given video (Section 3.6). Some artefacts are left during the video generation process, so we propose a postprocessing step to remove these (Section 3.7).

## 3.1 Monocular Reconstruction

Following similar 3DMM-based neural texture approaches (Thies et al., 2020; 2019), we first fit a 3DMM to each frame of the ground truth videos using differentiable rendering. For the 3DMM, we use FLAME (Li et al., 2017). FLAME uses a combination of linear blendshapes and blend skinning to control a full-face rig with 5023 vertices. We use a three-stage process for this fitting. We cover more detail in the supplementary.

**Stage 1:** First, we estimate the shape parameters of the FLAME model using MICA (Zielonka et al., 2022). MICA predicts the shape parameters from a single frame and is shown to be very accurate. We then fix the shape parameters. **Stage 2:** Now, with the shape fixed, we optimise other non-varying parameters by jointly optimising a regularised photometric loss function over several frames simultaneously. We then fix the texture and camera parameters in addition to the shape. **Stage 3:** Finally, we optimise the variable parameters for each frame. These are expression, pose and lighting. We initialise parameters for frame $t$ using the parameters for frame $t-1$.

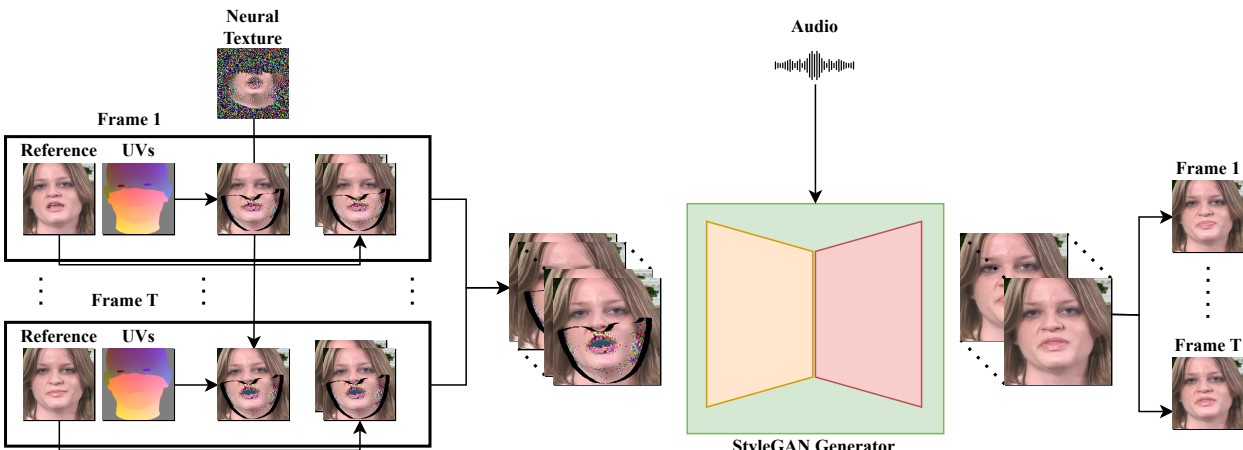

Figure 3: The architecture of our model. We take the UV rasterisations of each frame in a window of length T and sample a neural texture. We combine a mask with the real frame (Figure 4). These are concatenated with random reference frames of the same person and stacked across the channel dimension. A StyleGAN2-based (Guan et al., 2023; Karras et al., 2020) generator (see supplementary) is then used to convert these into T photorealistic frames.

Figure 4: The input to the image-to-image network. We sample the neural texture onto the rasterised mesh and use the mesh to estimate a mask. The input is computed using this mask, the target frame, and the rasterised texture.

## 3.2 Preprocessing

We next crop the face to $256 \times 256$ pixels. We find that pretrained face detectors ((Lugaresi et al., 2019; Deng et al., 2020)) suffer from two issues. The first is jitter between frames, and the second is the bounding box, which varies based on the jaw pose. We use our tracking data to generate bounding boxes to overcome these issues. We project the vertices of the meshes to obtain 2D landmarks using the parameters in Section 3.1. To prevent the jaw position from appearing in the box size, we project landmarks with the jaw in several positions and find the bounding box containing all these. We then add a small margin to get the final box. More details can be found in the supplementary material. We also use the tracked data to generate masks with the same multi-jaw approach. We rasterise a predefined mouth texture mask, which may be seen in Figure 4.

## 3.3 Architecture

In this section, we describe the architecture of our model. Inspired by previous work (Thies et al., 2019), we adopt a deferred neural rendering approach using neural textures. The model contains two components: learnable neural textures which are similar to standard RGB, uv-based, texture images with high-dimensional feature vectors; and a deferred neural renderer, an image-to-image network that takes these rasterised neural features and converts them into realistic video (Figure 3).

Existing ***person-specific*** models require several minutes of ***subject-specific*** training data. Our work's primary novelty is adapting this method to work few-shot, using only a small dataset of a given actor. A key insight to obtaining this is to note that much of the person-specific information can be stored in the neural textures, allowing the image-to-image network to be generalised across multiple subjects. To help the image-to-image network generalise, we use a reference frame as is done in person-generic works (Prajwal et al., 2020; Gupta et al., 2023; Guan et al., 2023; Shen et al., 2023; Stypulkowski et al., 2024).

To improve the quality of the generations, we replace the UNet used in previous works with a modified version of the StyleGAN2 (Karras et al., 2020) architecture **shown to be superior in** StyleSync (Guan

et al., 2023). Instead of providing masked target frames as is done in StyleSync, we mask out the target frame using the rasterization of a predefined mask on the 3DMM (Section 3.2) and fill in the masked regions with the rasterized neural texels. This can be seen in Figure 4.

Using the generator architecture from StyleSync also allows us to condition the video generation on audio. To improve temporal stability, we provide the generator with access to a window of frames surrounding the target and predict the same window of the final video, **as is shown effective in Wav2Lip (Prajwal et al., 2020)**. The architecture is best understood by viewing Figure 3 and referring to the supplementary material and StyleSync paper (Guan et al., 2023). In short, however, we convert input audio into MEL-spectrograms and use a series of 2D, residual convolutional layers to get a latent representation of audio, which is used in place of style vectors in the StyleGAN-based generator (Karras et al., 2020).

### 3.4 Training the Prior Model

We use multiple identities to train the prior deferred neural rendering network model. The network weights are shared for all identities, but we have a different, randomly initialised neural texture for each. We jointly optimise the network and textures to minimise the following loss, as it has been shown to be effective in previous work (Thies et al., 2020):

$$\mathcal{L} = \lambda_1 \mathcal{L}_1 + \lambda_{\mathrm{VGG}} \mathcal{L}_{\mathrm{VGG}} + \lambda_{\mathrm{reg}} \mathcal{L}_{\mathrm{reg}} + \lambda_{\mathrm{adv}} \mathcal{L}_{\mathrm{adv}} \tag{1}$$

$\mathcal{L}_1$ is a simple $\ell_1$ loss computed between the generated window of frames and the ground truth. To encourage the network to produce better results in the lower face and mouth region, we compute masks for these areas and weigh them higher. This mouth weighting has been shown to be effective in previous work, for example **FlashAvatar (Xiang et al., 2024)**. Specifically, we weigh the mouth region at ten times the background and the upper face and the lower face region at eight times the background.

$\mathcal{L}_{\mathrm{VGG}}$ is a VGG-based (Johnson et al., 2016) style loss. This is computed using a pre-trained VGG network and is calculated for each frame in the window individually, taking the mean. This is a perceptual loss and is known to improve image quality.

$\mathcal{L}_{\mathrm{adv}}$ is an adversarial loss. We jointly train the model with a discriminator and use an LSGAN (Mao et al., 2017) formulation for the adversarial loss. The discriminator is identical to the one used in StyleSync (Guan et al., 2023), but it is shown all frames in the window to encourage temporal consistency as demonstrated in previous works (Prajwal et al., 2020).

Finally, $\mathcal{L}_{\mathrm{reg}}$ is a regularisation loss for the neural textures. It is computed as the $\ell_1$ distance between the first three channels of the rasterised texture and the target frame. This encourages the first three channels of the texture to mimic a standard, diffuse RGB texture. $\lambda$ values are given in section 4.

### 3.5 Adapting to New Identity

Given a new actor, we adapt our model to them. We first initialise a new random neural texture and use the deferred neural rendering prior network from section 3.4. We train the texture from scratch but use the prior network as initialisation for the deferred neural renderer. To help the model maintain its generalisation, we also include data from other identities from the training set of the generic model. Specifically, we include person-generic training data in the person-specific dataset at a ratio of 1:1. We refer to this as a **mixed training strategy**. The mixed training strategy allows the deferred neural rendering network to continue learning from a comprehensive data distribution, including, for example, poses that may not be in the actor-specific dataset. This effectiveness is shown in table 4.

### 3.6 Audio-to-Expression Model

Given our neural rendering model, we can convert rasterizations of the 3DMM to realistic video. To change the lip motions of the video, we need to alter the model's expression parameters. To do this, we make use of state-of-the-art speech-driven animation models. In particular, we use Imitator (Thambiraja et al., 2023).

| Method | HDTF 100 Frames | | | | | | | HDTF 1000 Frames | | | | | | | CelebV-HQ (Average 200 frames) | | | | | | |
|---|---|---|---|---|---|---|---|---|---|---|---|---|---|---|---|---|---|---|---|---|---|
| | PSNR↑ | SSIM↑ | FID↓ | FVD↓ | Qual↑ | Lip↑ | ID↑ | PSNR↑ | SSIM↑ | FID↓ | FVD↓ | Qual↑ | Lip↑ | ID↑ | PSNR↑ | SSIM↑ | FID↓ | FVD↓ | Qual↑ | Lip↑ | ID↑ |
| Gupta et al. (2023) | 27.70 | 0.895 | 6.78 | 100.52 | 3.53 | 3.67 | 3.10 | 27.70 | 0.895 | 6.78 | 100.52 | 3.53 | 3.67 | 3.10 | 24.81 | 0.832 | 15.84 | 270.96 | 3.31 | 2.38 | 4.08 |
| StyleSync-G (Guan et al., 2023) | 29.26 | 0.899 | 7.07 | 99.10 | 3.77 | 3.50 | 3.20 | 29.26 | 0.899 | 7.07 | 99.10 | 3.77 | 3.50 | 3.20 | 29.14 | 0.895 | 9.26 | 285.75 | 3.23 | 2.77 | 3.38 |
| TalkLip (Wang et al., 2023a) | 28.34 | 0.887 | 9.98 | 128.53 | 2.47 | 3.59 | 3.03 | 28.34 | 0.887 | 9.98 | 128.53 | 2.47 | 3.59 | 3.03 | 28.85 | 0.896 | 13.89 | 346.01 | 2.92 | 2.85 | 3.54 |
| DiffDub (Liu et al., 2024) | 26.64 | 0.855 | 7.87 | 178.73 | 3.09 | 3.11 | 3.80 | 26.64 | 0.855 | 7.87 | 178.73 | 3.09 | 3.11 | 3.80 | 23.81 | 0.789 | 16.52 | 186.05 | X | Y | Z |
| MuseTalk (Zhang et al., 2024) | 28.74 | 0.899 | 5.95 | 119.78 | 2.80 | 3.03 | 3.57 | 28.74 | 0.899 | 5.95 | 119.78 | 2.80 | 3.03 | 3.57 | 25.01 | 0.848 | 14.90 | 202.69 | X | Y | Z |
| TalkLip-FT (Wang et al., 2023a) | 29.19 | 0.887 | 28.97 | 1218.89 | 2.57 | 3.06 | 3.60 | 27.95 | 0.864 | 10.21 | 126.95 | 2.69 | 2.86 | 3.57 | 28.42 | 0.870 | 25.97 | 254.04 | X | Y | Z |
| DiffDub-FT (Liu et al., 2024) | 26.71 | 0.857 | 7.69 | 154.12 | 3.17 | 3.31 | 3.69 | 26.77 | 0.858 | 7.73 | 166.04 | 3.29 | 3.23 | 3.80 | 23.84 | 0.799 | 18.120 | 192.86 | X | Y | Z |
| Ours Baseline | 27.92 | 0.888 | 10.52 | 138.06 | 2.90 | 3.33 | 3.13 | 29.00 | 0.899 | 8.61 | 126.45 | 3.06 | 3.23 | 3.10 | 26.61 | 0.870 | 16.29 | 187.2 | 2.31 | 3.23 | 3.23 |
| Ours Full | 29.10 | 0.899 | 5.76 | 99.10 | 3.57 | 3.77 | 3.83 | 29.30 | 0.904 | 5.44 | 94.161 | 3.90 | 3.80 | 4.03 | 30.17 | 0.912 | 5.35 | 86.65 | 3.38 | 3.69 | 4.46 |
| Real | 100.00 | 1.00 | 0.00 | 0.00 | 4.53 | 4.60 | 4.46 | 100.00 | 1.00 | 0.00 | 0.00 | 4.53 | 4.60 | 4.46 | 100.00 | 1.00 | 0.00 | 0.00 | 4.69 | 4.85 | 4.85 |

Table 1: Quantitative comparisons of our model with state-of-the-art. We compare in three settings, a very low data setting (100 frames) and a somewhat low data setting (1000 frames), both using HDTF and an unseen test set (CelebV-HQ). Our method is compared to **methods that see only one reference frame such as** TalkLip (Wang et al., 2023a), Gupta et al. (2023), StyleSync (Guan et al., 2023), DiffDub (Liu et al., 2024) and MuseTalk (Zhang et al., 2024) as well as **models that see multiple references frames** including a baseline version of our model trained from scratch as well as fine-tuned versions of TalkLip and DiffDub, given the -FT suffix. We compare using quantitative (*italics*) and user ratings (**bold**). We highlight the Best and Second Best for each metric (excluding the ground truth).

Imitator is a transformer-based model that allows for speaking style adaptation. We use the pre-trained Imitator model. To adapt to the speaking style of the new individual, we add a final layer to the network that independently applies a linear transformation for each expression and jaw pose parameter. We then train only this layer for each individual.

### 3.7 Post Processing

While our method produces high-quality results in the facial interior, it occasionally suffers from artefacts around the border between the face and background. Due to the strong bias introduced by the neural texture, pixels beyond the face region appear "stuck" to the face and follow its motion. This is best seen in video format (see the supplementary video). To reduce the effect of this artefact, we apply post-processing. We first apply a semantic segmentation network (Yu et al., 2021) to each generated and real frame. This separates the face and neck from the background. We can then replace the generated pixels with the real frame where both the generated and real frame agree the pixel is the background.

## 4 Implementation Details

To train our prior network, we used an Adam Optimiser with a learning rate of $1e-4$ and a batch size of 4. We set $\lambda_1 = 10.0$, $\lambda_{\mathrm{adv}} = 1.0$, $\lambda_{\mathrm{VGG}} = 10.0$, $\lambda_{\mathrm{reg}} = 5.0$. We have one neural texture for each identity; these have a $256 \times 256$ resolution and 16 channels. The audio encoder and Generator architecture are taken from StyleSync (Guan et al., 2023), but the first convolutional layer is altered to reflect the difference in the image channels of the input (e.g. StyleSync uses 3-channel RGB and we use 16-channel neural texture features, see the appendix). The prior network is trained for seven days using an NVIDIA V100 GPU. The fine-tuning stage requires only 1-2 hours of training on the smaller L4 GPU.

## 5 Results

**Data:** We train our prior model using the HDTF Zhang et al. (2021) dataset. HDTF consists of around 400 videos, each in high-definition and several minutes long. The length of the videos is important as we want to run experiments with various video lengths, as is done in section 5.6. We select a random subset of 20 videos for finetuning and testing, and the rest is used for the generic pretraining stage. We manually inspect the dataset to ensure that the subjects in the training set are not also accidentally included in the test set. We resample each video to 25fps and use 1500 frames (1 minute). We use the last 10 seconds as testing data and subsets of the remaining 50s for fine-tuning. In addition, our model can generalise beyond the dataset on which it is trained. To show this, we also include results from a subset of CelebV-HQ Zhu et al. (2022) (note that the prior model does not see this dataset). CelebV-HQ videos are not as long, so

TalkLip
Gupta et al.
StyleSync-G
DiffDub
MuseTalk
Ours (4s)
Ours (40s)
Real

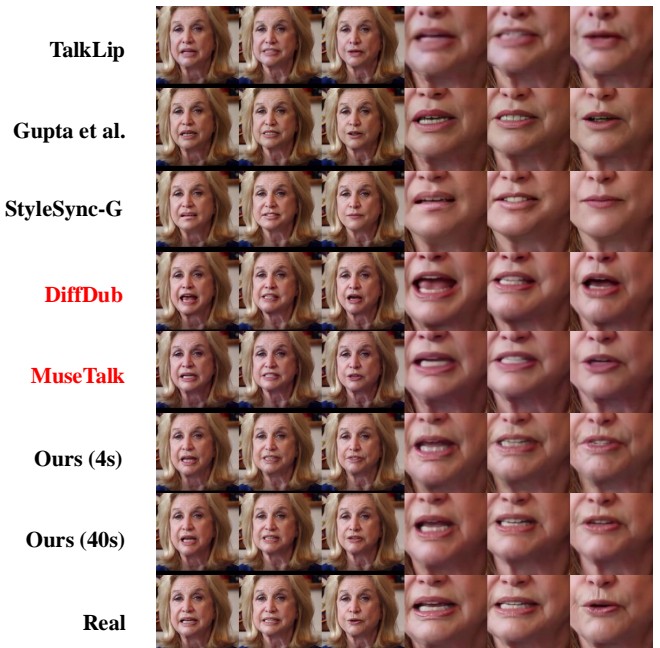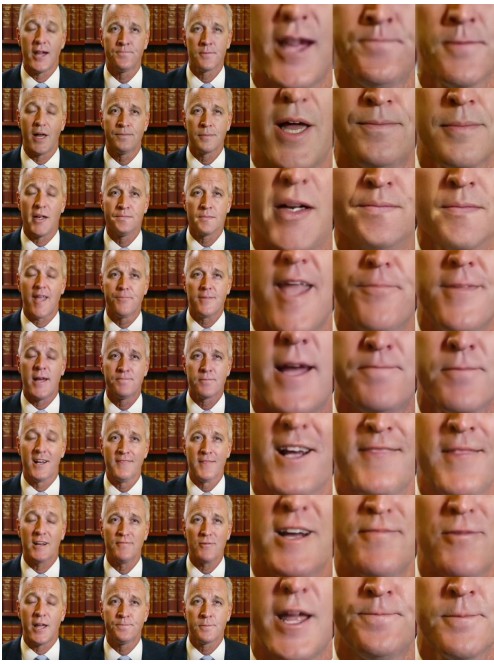

Figure 5: Qualitative comparisons to the state-of-the-art person generic models. We compare to TalkLip (Wang et al., 2023a), Gupta et al. (2023) and StyleSync (Guan et al., 2023). We show two versions of our model, one fine-tuned on 100 frames of data and one with 1000 frames. We also show the ground truth frames for comparison. Closeups of the mouth region are included for more detail.

we use the last 10 seconds for testing and all the remainder for fine-tuning. This is, on average, 200 frames. **We randomly select a further 20 videos from this dataset with the 'talking' annotation.**

**Metrics:** We look to evaluate our method on three criteria, **visual quality, lip sync** and **idiosyncracies**. Visual quality is measured using **FID at the image level and FVD at the video level**. As ground truth is available during re-enactment experiments, we also use **SSIM and PSNR**. While useful as proxies, these metrics are less important than how users perceive the method. For this reason, we also ask users to rate the three qualities: visual quality **(QUAL)**, idiosyncracies **(ID)** and lip-sync **(LIP)** out of 5. A total of 30 users provided ratings. Further details of this user study are provided in the supplementary material.

## 5.1 Comparisons to State-of-the-Art

We compare our model to the state-of-the-art. We separate these into person-specific and person-generic. To demonstrate the ability of our model to work for small and medium-sized datasets, we consider three scenarios: one with limited data (100 frames), one with significantly more data (1000 frames) and one using a different dataset (Zhu et al., 2022). We compare our model to three recent person-generic methods: **that of Gupta et al. (2023)**, which uses a VQ-GAN to achieve ultra-high resolution outputs; TalkLip (Wang et al., 2023a), which uses a lip reading network for better lip-sync the StyleGAN2 (Karras et al., 2020) based StyleSync (Guan et al., 2023) as well as the diffusion based DiffDub Liu et al. (2024) and MuseTalk Zhang et al. (2024). **For more detail on the baselines, please refer to the supplementary** We also compare our work with a baseline model. For this, we train our model on each subject from scratch. We consider this a close re-implementation of similar pipelines (Thies et al., 2020; 2019). Therefore, we do not also compare our work to these models.

The results are shown in Table 1. **Our model outperforms the person-generic models in terms of visual quality as measured by FID**. This effect is more noticeable with additional data. User ratings for quality are slightly lower for our model with just 100 frames but higher with 1000. However, **our model can**

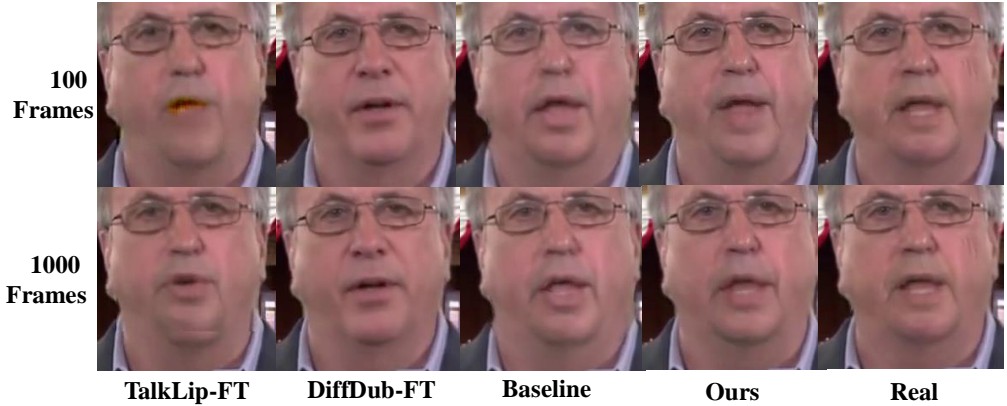

**100 Frames**

**1000 Frames**

| TalkLip-FT | DiffDub-FT | Baseline | Ours | Real |

Figure 6: **Qualitative comparisons to state-of-the-art person-specific models. We compare to fine-tuned versions of TalkLip (Wang et al., 2023a) and DiffDub (Liu et al., 2024), as well as a baseline model trained from scratch.**

| Row > Col % (95% CI) | Audio only | StyleSync-G | Baseline | Ours |
|---|---|---|---|---|
| Audio Only | - | **24** (16-33) | **23** (15-32) | **9** (4-16) |
| StyleSync | **76** (67-84) | - | **68** (58-77) | **38** (29-48) |
| Baseline | **77** (68-85) | **32** (23-42) | - | **9** (4-16) |
| Ours | **91** (84-96) | **62** (52-71) | **91** (84-96) | - |

Table 2: User study performed on a translated section of an in-the-wild video clip. We show the percentage of 35 users who preferred the row to the column. We include 95% confidence intervals for each **in brackets**.

**capture person-specific details that the generic models can not**. This is evidenced in Figure 5 and by the user ratings for idiosyncrasies (ID). This may suggest that the generic models are producing visually appealing but generic-looking lips. Note that we do not use LSE metrics (Prajwal et al., 2020) for lip sync as some previous works do. We find that the person-generic models optimise this directly to the extent that they outperform the ground truth (9.34 for StyleSync vs 7.28 for real video), making this an unreliable metric. To further this point, users preferred the lip-sync of Gupta et al. (Rating=3.67) to StyleSync (3.50), but StyleSync outperforms it using LSE-C (9.34 vs 7.00), and this is more pronounced with real data (User Ratings = 4.6, LSE-C=7.28). Instead, we believe that the user experience of lip sync is the most important, and our model outperforms all others in this respect.

Compared to person-specific models, our method outperforms them across all metrics. **The difference is most prominent when trained on 4 seconds (100 Frames) of data, suggesting our model is using the available data effectively**. We further investigate this effect in Section 5.6. The NeRF-based models, in particular, fail with unseen poses when trained on just 100 frames. This can be seen easily in Figure 6.

## 5.2 User Study

To investigate our model in its intended context, altering the lip motion to match dubbed audio in a different language with limited data, we design a user study to replicate this. We take three videos of politicians speaking in their native language and use the automated (audio-only) dubbing provided by 11labs (Ele). These video clips are 15-20 seconds long, much shorter than those used in previous works (Thies et al., 2020; Wen et al., 2020; Ye et al., 2023). We compare our work to the highest-quality generic and specific models, measured by user ratings. We also consider audio-only dubbing (not altering the lips), which remains the industry standard. We perform a forced choice experiment. Users are given the same video dubbed using two random methods from our selection. The users are asked which they prefer. The results are in Table 2 and show that users prefer our method to all others within a 95% confidence interval. Further details of the user study are outlined in the supplementary material.

| Method | \multicolumn{6}{c}{Train Iterations to Reach PSNR} |
|---|---|---|---|---|---|---|
| | 25 | 26 | 27 | 28 | 29 | 30 |
| Baseline | 2200 | 3000 | 6200 | 9800 | 18000 | 40000 |
| Ours | 200 | 300 | 300 | 400 | 700 | 1600 |
| Speedup Factor | 11 | 10 | 20.7 | 24.5 | 25.7 | 25 |

Table 3: Number of iterations (to the nearest 100) taken to reach a given PSNR value for our model and the baseline. Average of three runs with different identities.

| Method | \multicolumn{7}{c}{HDTF 100 Frames} |
|---|---|---|---|---|---|---|---|
| | PSNR ↑ | SSIM ↑ | FID ↓ | *FVD* ↓ | **Qual** ↑ | **Lip** ↑ | **ID** ↑ |
| Ours w/o post-processing | 28.20 | 0.888 | 5.28 | 98.10 | 3.20 | 3.50 | 3.33 |
| Ours w/o mixed data | 28.95 | 0.897 | 5.88 | 107.30 | 3.17 | 3.50 | 3.33 |
| **Ours full** | 29.10 | 0.899 | 5.76 | 99.10 | 3.57 | 3.77 | 3.83 |

Table 4: Results of the ablation study. Including our post-processing step (Section 3.7) and mixed training strategy (Section 3.5) improves the results across many metrics. We highlight the best results .

### 5.3 Ablations

We perform an ablation study of our model. We show that the post-processing step (Section 3.7) and the mixed training strategy for fine-tuning (Section 3.5) both improve the results of our model. We use the 100-frame setting for this experiment. The results can be seen in table 4. The mixed training strategy improves results across all metrics, showing that it helps the model generalise. The post-processing increases FID, suggesting worse visual quality. However, the user ratings show a preference for post-processing. This may be because the post-processing noticeably removes the sharp boundary between the real and generated frame at the cost of some reconstruction accuracy.

### 5.4 Training Speed

Our model trains faster **than a baseline model trained without a prior**. As this prior network works across many identities, adapting to a new identity requires much less training. To demonstrate this effect, we compare our model to the baseline. This model is trained from scratch for each new identity, which does not use a prior network. We show that our model is faster to train by recording the number of training iterations required to reach a given value of PSNR on a withheld validation set. To show that this effect is not just due to limited data making the baseline weaker, we train on the 1000 frame setting. The results are shown in Table 3 and show clearly that **our model converges an order of magnitude faster than the baseline**. This takes 1-2 hours on an L4 GPU.

### 5.5 Inference Speed

Our prior is not required at inference time, as it is only used to train the model. The additional steps we require for audio-to-expression generation and rasterization are very fast compared with the heavy image-to-image generator network used in these methods. This means that our model runs close to the same inference speed as image-to-image models (e.g. Thies et al. (2020); Guan et al. (2023); Gupta et al. (2023)) In practice, **excluding reconstruction,**, we can generate frames at around 5fps on an $L4$ GPU without any specific code optimisation. **Including reconstruction, our model is closer to 0.2fps. Which is slower than baselines, which run at between 5 and 30fps. However, this only needs to be computed once per video, which can then be dubbed in many languages at the faster 5fps.**

**Baseline**

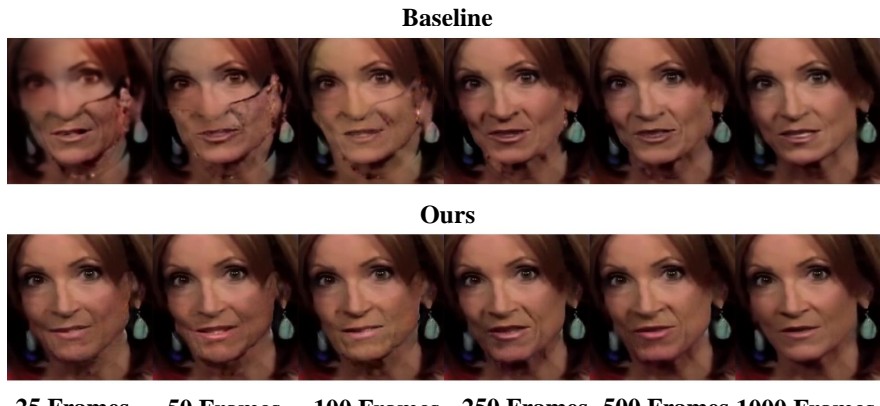

**Ours**

**25 Frames    50 Frames    100 Frames    250 Frames  500 Frames  1000 Frames**

Figure 7: The effect training dataset size. The baseline model is trained from scratch and suffers when using limited data. Ours, by comparison, is far more robust.

| N Frames | 100 | 250 | 500 | 1000 |
|---|---|---|---|---|
| Baseline | 17.71 | 12.33 | 11.84 | 11.78 |
| Ours | 11.37 | 11.88 | 11.74 | 11.21 |

Table 5: Our model is robust even on datasets **as small as four seconds (100 frames)**. We compare the number of frames of a given actor to the FID obtained by the model trained on this dataset.

### 5.6 Effect of Dataset Size

Our method allows dubbing actors with only a few seconds of data. To demonstrate this ability, we compare our prior network method to a baseline model, which trains both the deferred neural rendering network and neural texture from scratch. We evaluate the model using 10-second clips of one of our target actors, having trained both models on subsets of the training data of various sizes. The results are shown in Table 5, Figure 7 and in the supplementary video. It can be seen that **our model produces much higher quality results than the baseline on the small datasets**, but this effect reduces with dataset size.

### 6 Limitations and Future Work

While our method achieves state-of-the-art, is robust to small datasets and trains fast, it is not without limitations. First, there are noticeable artefacts around the border between the face and the background. The post-processing we introduce does mitigate this, but not completely. We think this could be addressed by first segmenting the background from the person, training the model on the foreground only and composing the result. We will address this in future work. Another significant issue is that the monocular reconstruction stage is very slow as it relies on optimisation through a differentiable renderer. Recent work has shown (Feng et al., 2021; Danecek et al., 2022; Filntisis et al., 2022) that regression-based reconstruction is possible, but it is still not temporally consistent enough for our purposes. We would like to investigate temporal regression models to this end, that could run in real-time. **It is also a limitation of our work that it requires at least a few seconds of data to produce good results and performs poorly with only a single frame, while person-generic models are able to effectively use this amount of data.**

## 7 Ethical Discussion

It is of paramount importance that the benefits and harms of the field of audio-driven visual dubbing are correctly weighted. This section discusses these and details our attempts to mitigate any harm.

**Privacy:** The HDTF dataset is released under the CC BY 4.0 License. We, therefore, have permission from the authors to use this dataset. To help protect the privacy of the individuals in this dataset and comply with GDPR, we will provide a contact form allowing any individuals to remove themselves from the dataset and model. As our model is two-part, consisting of neural textures and a generic rendering network, the model cannot reconstruct an individual without their neural texture. Simply deleting the texture will ensure that the individual is no longer represented in the model.

**Associated Harms:** The potential for misusing our technology is significant. 'Deepfakes" refers broadly to the class of artificially generated videos of which our work may be considered. These models are known to cause harm through misinformation, defamation and non-consensual explicit material. To help mitigate these harms, we will only provide access to the model to researchers at an accredited institution. Furthermore, we are investigating invisible watermarking (Navas et al., 2008; Bui et al., 2023) and deepfake detection methods (Rössler et al., 2019; Mirsky and Lee, 2021).

**Associated Benefits:** Our model enables media to cross language barriers. This helps promote diverse societies and allows for the spread of various cultures. In addition to this, the method has significant potential economic value. In these ways, the development of such models can benefit societies.

The exact weighting of the good and harms of developing "deepfake" models remains an open question. Still, we believe that visual dubbing, with its potential for spreading culture and the economic benefits, outweigh the potential risks when considering the mitigation we have put in place.

## 8 Conclusion

We have presented Dubbing for Everyone. Unlike person-specific models, which require several minutes of personalised data, or person-generic models, which cannot capture personalised appearances, our model is a hybrid person-generic, person-specific model. Using adaptation, our model is capable of high-quality, personalised visual dubbing using a few seconds of data for a given actor. Our experiments have shown that our model archives state-of-the-art across many metrics, including user ratings. We have also shown that our person-generic prior network training and adaptation strategy **trains faster, reaches higher quality and works on less data** than **an indentical** baseline model trained without priors.

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

# A  Further Methodological Details

## A.1  Monocular Reconstruction

Monocular reconstruction is the process of fitting a 3D mesh to the video. We follow a similar pipeline to existing works (Zielonka et al., 2022; Thies et al., 2016; 2020; Grassal et al., 2022). We use differentiable rendering using PyTorch3D to optimise a set of parameters to fit the given video best. We use the FLAME model (Li et al., 2017) to generate 5023 vertices from a set of parameters:

$$V = F(\alpha, \beta, \theta) \tag{2}$$

Where $\alpha \in \mathbb{R}^{300}$ are the parameters for shape, $\theta \in \mathbb{R}^{100}$ the expression parameters and $\phi$ the pose parameters for jaw, neck rotation and translation. We then model the image formation process using a perspective camera with intrinsic $K$ and extrinsic $R$, the PCA-based FLAME texture model with parameters $\beta$, and lighting (assumed distant and diffuse) using 9-band spherical harmonics with parameters $\gamma$. We optimise subsets of these parameters based on the following loss:

$$\mathcal{L} = \lambda_{\text{photo}}\mathcal{L}_{\text{photo}} + \lambda_{\text{land}}\mathcal{L}_{\text{land}} + \lambda_{\text{reg}}\mathcal{L}_{\text{reg}} \tag{3}$$

Where $\mathcal{L}_{\text{photo}}$ is the pixel-space $\ell_1$ loss between the rasterised image and the real frame. $\mathcal{L}_{\text{photo}}$ is the $\ell_2$ distance between the projection of manually labelled landmarks on the FLAME mesh and corresponding landmarks detected with Mediapipe (Lugaresi et al., 2019). $\mathcal{L}_{\text{reg}}$ is a regularisation loss based on the $\ell_2$-norm of the shape, expression and pose parameters.

In stage 1, we predict $\alpha$ using the pre-trained MICA (Zielonka et al., 2022) model and do not change these after. In stage 2, we optimise $\mathcal{L}$ with respect to all parameters except $\alpha$ across several randomly selected frames. We then leave $\alpha, \beta, K, R$ all fixed and optimise $\mathcal{L}$ with respect to $\theta$ and $\phi$ for each frame sequentially starting at frame 0. $\theta$ and $\phi$ are initialised at frame $t$ using their values at frame $t-1$. All optimisation is done with the Adam optimiser.

## A.2  Preprocessing

To get a more stable bounding box for our preprocessing, we use the monocular reconstruction results. As the reconstruction is temporal and uses photometric losses, it is much more stable than bounding box estimation. To get a bounding box from the reconstructed mesh naively is simple. We produce the posed 3D mesh from the parameters using the FLAME model. We can then project the vertices of this mesh onto the image plane using the parameterised camera and take a bounding box that contains all the vertices plus some margin.

Unfortunately, both this method and existing bounding box detectors have an issue with the jaw's position. The bounding box is longer when the jaw is open than when it is closed. This may be desirable for some applications but is detrimental to our purposes. The reason is that the bounding box can bias the generation process. If the generator sees a longer box, it will infer that the jaw is open, so it will perform less well when trying to convert a frame with an open jaw to a closed one. We overcome this by making our bounding box jaw-independent. We set the jaw parameter of the FLAME model to fully closed, keeping everything else the same, and project the vertices. Then, we set it to fully open and do the same. We can then take the union of these points and get the bounding box from these.

## A.3  Prior Network Architecture

Our prior network architecture is similar to StyleSync (Guan et al., 2023). In particular, we use two encoders: one for the face and one for the audio. A StyleGAN2 (Karras et al., 2020) decoder is used to generate the final frames. This decoder starts from a latent space, denoted as the W space in the StyleGAN literature. It uses a series of upsampling and convolutional layers modulated by the style vector W. We use the encoded

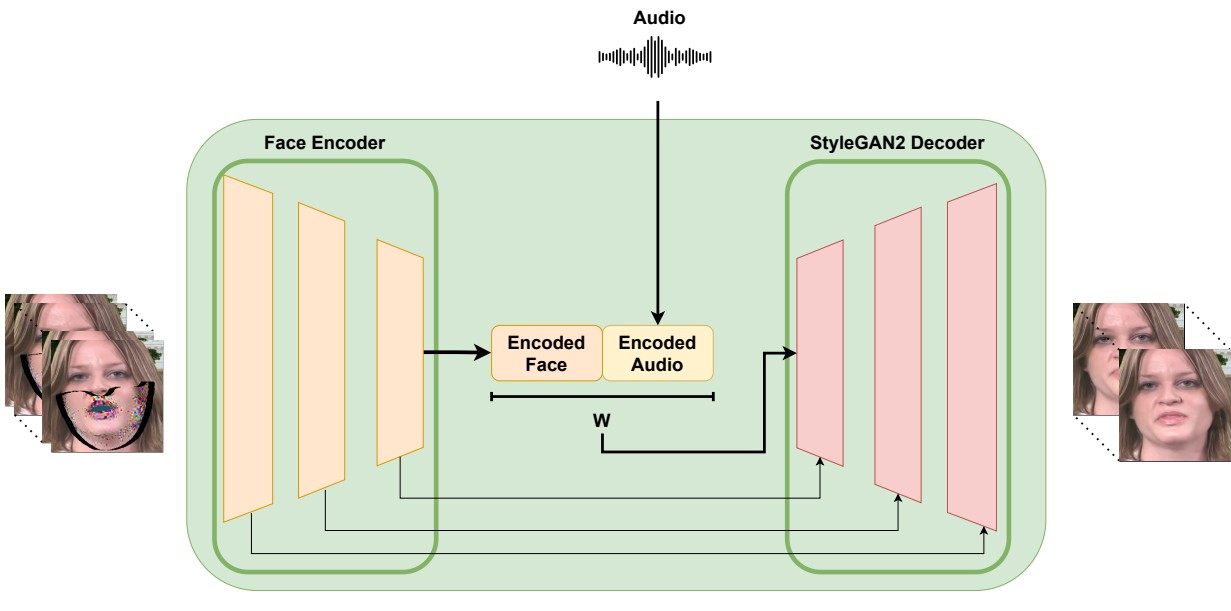

Figure 8: The architecture of our StyleSync (Guan et al., 2023) based generator (see fig. 3. Both audio and rasterized face images are encoded and concatenated into a latent style vector, W. This is used to generate the final frames. Skip connections between the face encoder and the decoder are included to help the network better use the rasterized textures.

audio and face representations as our W vector. We also include skip connections between layers of the face encoder and the StyleGAN2 decoder to help the network best use the rasterized neural texture. This is shown in Figure 8. The audio encoder is a simple CNN and is the same as is used in several previous works (e.g. (Guan et al., 2023; Prajwal et al., 2020)). The face encoder is similar to the StyleSync version but has more input channels to handle the higher dimensional neural features.

## B   Further Results

### B.1   Two-Alternative Forced Choice User Study

The Two-Alternative Forced Choice User Study (2AFC) is the choice of experiment in Table 2. For this experiment, we show the effectiveness of our method in the target scenario. We take a short video clip of approximately 15 seconds and use an online service to translate the audio with voice cloning. We do this for three videos from speeches in various languages from the European Central Bank YouTube page (EUB).

We compare four methods. For our method, we finetune our pre-trained model on just 15 seconds of data. For the baseline model, we train the same model but from scratch, with all weights reinitialised. As StyleSync (Guan et al., 2023) scores best among the quantitative metrics besides our method, we use it for comparison. Finally, to show that altering the lips is necessary, we also consider the audio-only approach, where we keep the original video but change the audio stream.

To find out user preferences, we first show the user the real video and ask them to pay attention to the speaker's speaking style and appearance. We then show two side-by-side, dubbed videos of the same person. These are selected randomly from all four methods, and the order (left or right) is also random. We use Amazon's Mechanical Turk to collect these responses.

In total, 35 users completed our user study. 202 responses were recorded, meaning each user responded to an average of 5.8 comparisons. Each response was for all three videos. The vast majority (91%) of users preferred our method to audio-only dubbing, suggesting that our method outperforms the industry standard.

A similar percentage (91%) preferred our method to the baseline. This is likely due to the relatively small size of the training dataset. Compared with StyleSync (Guan et al., 2023), a smaller but still significant majority (61%) preferred our work, suggesting that our model is indeed state-of-the-art for visual dubbing.

We computed 95% confidence intervals in table 2 using the assumption of a Binomial random variable for if users prefer method X to method Y. We treat each comparison for each of the three videos as a separate sample.

### B.2 Ratings User Study

We also conduct a ratings user study to compare our method to the existing state-of-the-art. There are three qualities we wish to measure. These are visual quality, which we denote QUAL in the results tables; idiosyncrasies, which we denote ID; and lip-sync, which we denote LIP. For each video and method, we ask users to rate these three qualities out of five, where one is very poor, and five is very good. For visual quality, we ask users to consider artefacts and blur. For lip-sync, we ask users to consider how well the lip movements match the audio. Finally, with idiosyncrasies, we ask users to consider how much the video looks like the reference video, that is, the actual video being reconstructed, in terms of facial, lip, mouth and teeth appearance and motion. The subject video and the reference are played separately to prevent poor lip-sync from affecting the perceived idiosyncratic quality.

Results are collected again using Mechanical Turk. In this case, users are asked to rate all available methods in the same session, using the same subject. This allows the user to calibrate their ratings relative to all available methods. To prevent bias, the selected subject and the order in which the methods are shown to the user are randomised for every user. In total, 30 users completed the study, with each rating all methods exactly once. We report the mean opinion score in Table 1 and Table 4

## C Further Comparisons

### C.1 Further Quantitative Comparisons

In this section, we include additional Quantitative comparisons with recent state of the art, including DiffDub (Liu et al., 2024), MuseTalk Zhang et al. (2024). We also compare with additional person-specific baselines in the form of fine-tuned versions of TalkLip Wang et al. (2023a) and DiffDub Liu et al. (2024). In both cases, we use the official training code, with all the same losses and hyperparameters. We begin the fine-tuning from the checkpoints officially released. We show these results in table 1 and show that our model still achieves state-of-the-art.

### C.2 Qualitative Visualization of Post-Processing

We show qualitatively the results of the post-processing algorithm in fig. 9. It can be seen that artefacts on the border between the face and the background are removed. This effect is most noticeable temporally, and can be seen in the supplementary video.

### C.3 Pose Robustness

In fig. 10, we show how our model is able to remain robust under common pose variation. We use videos from the CelebV-HQ (Zhu et al., 2022) to demonstrate this, where the pose variation is higher than in HDTF. Despite our prior being trained on HDTF only, with minimal pose variation, we find our model is robust to the amount of pose variation typical in in-the-wild videos, such as CelebV-HQ.

### C.4 Further Details on Baselines and Datasets

Here, we discuss further details about each of the baselines we compare with:

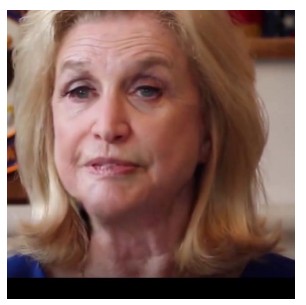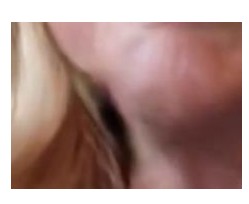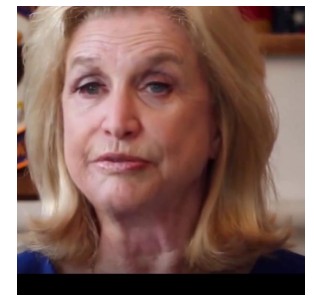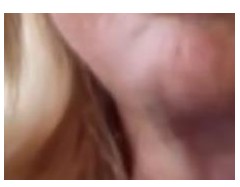

**Without Post Processing**          **With Post Processing**

Figure 9: **An example of the effect of the post processing step in reducing artefacts on the border between the face and background. We show a zoomed in part of this region to demonstrate this effect.**

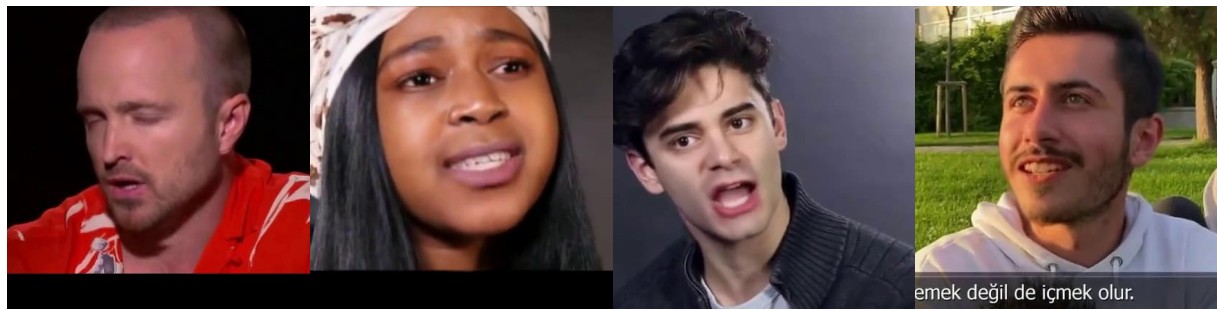

Figure 10: **Our model handles pose variation of around 30 degrees in all dimensions well, as shown here.**

**Gupta et al. (2023):** This method involves the use of a VQ-GAN (Esser et al., 2020) to produce a quantised, discrete latent space on which a generator and lip-sync discriminator operate. The model is trained on a 4K Talking Face dataset collected by the authors. It only sees one reference frame. We obtained our results by contacting the authors directly.

**StyleSync (Guan et al., 2023):** This paper uses a modified version of StyleGAN2 Karras et al. (2020) to perform lip generation from audio. It is trained on LRW and VoxCeleb2. There are two models outlined in this paper. StyleSync-G is a generic model that sees only one reference frame, and StyleSync-P is fine-tuned on a small number of frames. We contacted the authors for these results but could only obtain them for StyleSync-G, which we report.

**TalkLip Wang et al. (2023a):** TalkLip uses an expert lip-reading network to produce accurate lip-sync. It is trained on LRS2 and LRW. We use the officially released evaluation code and model checkpoint (Tal). Using the official training code, we can also fine-tune on person-specific data. We use the same loss formulation and hyperparameters as the training code. This is denoted TalkLip-FT.

**DiffDub Liu et al. (2024):** DiffDub is based on a diffusion autoencoder, which is conditioned on audio to inpaint the masked mouth region. It is trained on HDTF, like our model. We use the officially released code (Dif) and model checkpoint for the base model and also fine-tune using the training code released. This finetuned model is referred to as DiffDub-FT.

**MuseTalk Zhang et al. (2024):** MuseTalk is another diffusion method based on stable diffusion Rombach et al. (2021). This model is also trained on HDTF, like ours. We use the officially released inference code and model checkpoint. However, as no training code is provided, we do not compare with a fine-tuned version of this model.

Here, we discuss further details about each of dataset used for training and/or evaluation of the above models:

**HDTF:** Zhang et al. (2021) Is a high quality (720 or 1080p videos) dataset of 362 videos totalling 15.8 hours. It is collected from mostly political speeches on YouTube.

