# OpenReview forum: "Dubbing for Everyone: Cost and Data-Efficient Visual Dubbing using Neural Rendering Priors"
_TMLR — Rejected by TMLR_

### Review · Reviewer_Msom · 2024-12-20

**Summary Of Contributions:**

The authors propose a method for visual dubbing -- the task of inpainting the lower half of a face in a video, conditioned on an audio sample. Unlike some previous approaches, the authors train their method on both a multi-subject video corpus and footage of the test subject, which they describe as the work's "primary novelty." They evaluate reconstruction quality and qualitative metrics and find that their method outperforms zero-shot baselines. To evaluate dubbing of new audio, where reconstruction metrics cannot be applied, the authors perform a perceptual study with foreign language.

**Audience:**

Yes

**Broader Impact Concerns:**

The nature of the work entails serious broader-impacts concerns, and the authors describe misuse potential as "significant." The author's statement that "Simply deleting the texture will ensure that the individual is no longer represented in the model," is confusing as it appears the data would still be contained in the "multi-person prior network," making complete removal of an individual’s data more complex than suggested. It is also unclear how the authors have determined that their work "cannot be used for [the production of non-consensual explicit material]," as a supplied audio sample can obviously be explicit in nature, and the authors are requested to acknowledge this. The authors have stated their intention to restrict access to the model, but such restriction may also hinder reproducibility of the work.

**Claims And Evidence:**

No

**Requested Changes:**

In addition to the weaknesses outlined above, the authors are requested to address concerns relating to 1) claims; 2) discussion of related work; and 3) evaluation methodology. The authors are also recommended to review points of form in 4) to improve the clarity and accessibility of the paper.
## Claims
1. The claim of applicability to "background actors" is not supported by the evaluations. The evaluation datasets, HDTF (Zhang et al., 2021) and CelebV-HQ (Zhu et al., 2022), are self-described respectively as "high-definition" and "high-quality" and feature relatively constrained pose. The claim must be removed unless it is explicitly evaluated.
2. The statement that "Existing models require several minutes of [subject-specific] training data." is misleading, as most visual-dubbing methods are conditioned on a single frame.
3. The authors claim "Our model trains faster than existing person-specific models. This is due to the deferred neural rendering priors," but it must be clarified that the comparison is in reference to NeRF-based methods, so it is not appropriate to reduce methodological differences to use of "neural rendering priors."
4. The claim "We have also shown that our person-generic prior network training and adaptation strategy trains faster, reaches higher quality and works on less data than a similar model trained without priors" is misleading as it refers to an ablation study rather than comparisons with  a published method.
5. It has not been demonstrated that the model produces results that are "recognisable for any actor." The claim should be removed.
6. In claiming the proposed method is "an order of magnitude cheaper," the authors must clarify that they are referring to what they call "person-specific" methods, not those conditioned on a single frame, which are presumably orders of magnitude faster than the proposed method’s two-hour subject-specific finetuning requirement.
## Related Work
1. The related work section needs clearer task-specific organization. Many cited works address distinct tasks like avatar generation or image animation rather than visual dubbing, conflicting with the authors' characterization of them. The authors should distinguish between visual-dubbing works and those that solve distinct tasks.
2. The current related-work section organization (2.1 Person Generic Models, 2.2 Person Specific Models, and 2.3 Prior Learning for Faces) is confusing and could be more logically restructured as “Methods using single identity frames” and “Methods using multiple identity frames.” Section 2.3 appears redundant as most visual-dubbing methods in 2.2 (such as Kim et al. (2019); Song et al. (2020); and Thies et al. (2020)) can also be described as having "adopted a similar strategy of pretraining a prior model and fine-tuning to new identities."
3. Of related work, the authors claim "They all produce high-quality output but come with significant data requirements ranging from about 15 seconds Shen et al. (2022) to upwards of 10 minutes Du et al. (2023). In contrast, our method achieves similar quality using as little as 4 seconds of training data, thanks to our person-generic prior network training and person-specific adaptation." However, neither of the works referenced are compared against, nor are they visual-dubbing methods. The author should not make efficiency claims in relation to works intended to solve distinct tasks.
4. The authors' characterization of methods conditioned on a single image as lacking in quality by nature of them being "zero-shot" ("However, existing models are either zero-shot and, therefore, lack quality") was not demonstrated and should be removed.
5. The authors' claim that subject-specific methods require "off-putting user enrollment" is not supported by citations. Which related methods require such?
## Evaluation
1. Gupta et al. (2023) is not named Wav2LipHQ, which seems to refer only to an unrelated GitHub repository [5/A]. The authors should verify they are comparing against the intended method.
2. CelebV-HQ (Zhu et al., 2022) does not appear to be annotated as a speaking dataset. Would the authors please clarify how they selected clips for evaluation?
3. The authors write their method "can generate frames at around 5fps". Does this metric include application of the MICA (Zielonka et al., 2022) face tracker which is used during preprocessing? If accounted for, it seems the framerate should be closer to 0.2FPS. Would the authors please clarify what is included in the calculation?
4. The authors introduce a number of changes over previous work, such as 1) "To improve the quality of the generations, we replace the UNET used in previous works with a modified version of the StyleGAN2"; 2) "To improve temporal stability, we provide the generator with access to a window of frames surrounding the target and predict the same window of the final video"; or 3) "To encourage the network to produce better results in the lower face and mouth region, we compute masks for these areas and weigh them higher," but do not perform ablation studies. The effects of these changes should be explicitly evaluated.
5. To establish a fair comparison with person-generic methods, the authors should additionally evaluate with single-frame conditioning.
## Form
### Bibliography and Citations
1. The bibliography is not consistently formatted and includes a paper with no title (Khakhulin et al.). The authors should review the entries to ensure they are complete.
2. Bibliography entries for arXiv papers that have since been published, such as Du et al. (2023), should be updated.
3. A number of citations in the appendix appear as question marks and require correcting.
4. Motivational claims such as visual dubbing "allowing content creators to reach more viewers worldwide" require supporting citations.
### Formatting and Typos
1. The text alternates between British and American English, and should be standardized throughout.
2. The document is unnecessarily large due to the inclusion of a number of big or uncompressed images. To improve accessibility, the authors should properly compress their figures, and use vectorized versions whenever possible.
3. The following sentence flips the terms "person-specific" and "person-generic": "Our method is compared to person-specific models TalkLip (Wang et al., 2023a), Wav2LipHQ (Gupta et al., 2023) and StyleSync (Guan et al., 2023), and person-generic models including a baseline version of our model trained from scratch as well as GeneFace (Ye et al., 2023) and RAD-NeRF (Tang et al., 2022)." The authors should revise the sentence and verify other uses.
4. The authors should consider including the appendix at the end of the main PDF for accessibility. See "Format" in the author guidelines [6].
### Tables and Figures
1. The highlighting in Table 1 is not consistent, and the same color should be applied for ties. The coloring of "Lip" and "ID" in the middle section should also be explained.
2. The authors should consider adding color-coding to Table 2 to improve readability.
3. The authors write they "computed statistical significance in table 2," but they do not appear to have included the results.
4. The information in Table 3 would likely have been better conveyed in a line plot. The authors should consider revising it.
5. Figure 8 in the appendix appears to be missing a symbol, as there is nothing at the base of a directed arrow.

[5] https://github.com/lodelabuelo/wav2lipHQ

[5A] https://archive.ph/OIsRe

[6] https://jmlr.org/tmlr/author-guide.html

**Strengths And Weaknesses:**

## Strengths:
1. The model outputs included in the supplementary video are visually appealing.
2. The text is generally well-written.
3. A number of method comparisons are performed.
## Weaknesses:
1. The claimed "primary novelty" appears to be predicated on a misunderstanding of prior work ("Our key insight is to train a large, multi-person prior network, which can then be adapted to new users"; "The critical insight of our work is the realisation that existing high-quality models can benefit from a large-scale pre-training of some of their components"). The authors write they consider their method a "close re-implementation of similar pipelines (Thies et al., 2020; 2019) [when trained on only the target subject]." However, it must be clarified that Thies et al. (2020) self-describe their approach as similarly having “two stages -- the generalization and the specialization phase. In the first phase, [they] optimize for the shared network parameters that enable a generalization among different source actors." The authors must properly contextualize their work relative to prior efforts.
2. Relatedly, it is not clear what data was used to train the baseline models. If the baselines are not trained on both the same pre-training corpus and subject-specific samples, references to data efficiency must be removed.
3. The authors' comparisons against full avatar-generation methods, RAD-NeRF (Tang et al., 2022) and GeneFace (Ye et al., 2023), are not sound. These are not video-dubbing works, solving a distinct non-reconstruction task, and should not be evaluated using reconstruction metrics like PSNR and SSIM. The claim that the proposed method "generalizes to limited data better than existing person-specific models" is unsupported because no such models were included in the evaluation.
4. In the supplementary video, the authors claim they "compare to the most up-to-date state-of-the-art in the field." However, it should be noted that they make no reference to video-dubbing works published in the past year, such as [1], [2], [3], and [4]. Claims relating to "state-of-the-art" must be removed if the authors do not evaluate their proposed model against recent, published methods.
5. It is not clear why the authors have chosen their own evaluation setting rather than testing their method on a standard benchmark. Neither of the employed datasets were evaluated on in any of the baselines employed. The work would be stronger if evaluated in an existing setting against published results.

[1] Lee, Dongyeun, et al. "RADIO: Reference-Agnostic Dubbing Video Synthesis." Proceedings of the IEEE/CVF Winter Conference on Applications of Computer Vision. 2024.

[2] Liu, Tao, et al. "DiffDub: Person-Generic Visual Dubbing Using Inpainting Renderer with Diffusion Auto-Encoder." ICASSP 2024 IEEE International Conference on Acoustics, Speech and Signal Processing (ICASSP). IEEE, 2024.

[3] Liu, Kangwei, Xiaowei Yi, and Xianfeng Zhao. "ProDub: Progressive Growing of Facial Dubbing Networks for Enhanced Lip Sync and Fidelity." 2024 IEEE International Conference on Multimedia and Expo (ICME). IEEE, 2024.

[4] Zhang, Longhao, et al. "PersonaTalk: Bring Attention to Your Persona in Visual Dubbing." SIGGRAPH Asia 2024 Conference Papers. 2024.

---

> ### Author Response · Authors · 2025-01-16
> **Reply to Reviwer Msom**
>
> **We would like to thank the reviewer for this thoughtful and insightful feedback. We will address comments here point-by-point.**
>
> The claimed "primary novelty" appears to be predicated on a misunderstanding of prior work […] The authors must properly contextualize their work relative to prior efforts.
>
> **The word of Thies et al. does consider a pre-trained network. However, this is for the much easier task of the audio-to-expression component only, which we do not claim as a contribution. Thies et al. explicitly train one rendering network per subject. See, for example, page 2 of their work, “As a visual basis; we leverage a short target video of a real person…”, and “In our experiments, the target videos are comparably short (2-3 min)”. We will clarify that we are referring to the rendering part of such a pipeline only (Section 2.2 at end of the second paragraph).**
>
> Relatedly, it is not clear what data was used to train the baseline models. If the baselines are not trained on both the same pre-training corpus and subject-specific samples, references to data efficiency must be removed.
>
> **The baseline person-specific models are not trained on the pre-training data and are not designed to do so. They are designed for a single subject, as they learn a texture approximating appearance. This would not work for multiple subjects with differing appearances.
> For finetuning person-generic models on the personalised data, please see EXPERIMENT 2 in the shared reply.**
>
> The authors' comparisons against full avatar-generation methods, RAD-NeRF (Tang et al., 2022) and GeneFace (Ye et al., 2023), are not sound. These are not video-dubbing works, solving a distinct non-reconstruction task, and should not be evaluated using reconstruction metrics like PSNR and SSIM. The claim that the proposed method "generalizes to limited data better than existing person-specific models" is unsupported because no such models were included in the evaluation.
>
> **We believe these are fair comparisons as video-dubbing works. These methods directly compare to wav2lip in their own evaluation, showing that they intend their works to be used in this setting. To better support our claim, we also include finetuned versions of other models, as extra person-specific models. See EXPERIMENT 2.**
>
> The supplementary video claims that the authors "compare to the most up-to-date state-of-the-art in the field." However, it should be noted that they make no reference to video-dubbing works published in the past year, such as [1], [2], [3], and [4]. Claims relating to "state-of-the-art" must be removed if the authors do not evaluate their proposed model against recent, published methods.
>
> **Thank you for sharing these works. Of these, only DiffDub [2], provides code. To make up for this, we also include results from MuseTalk, also released in 2024. Please see EXPERIMENT 1.**
>
> It is not clear why the authors have chosen their own evaluation setting rather than testing their method on a standard benchmark. Neither of the employed datasets were evaluated on in any of the baselines employed. The work would be stronger if evaluated in an existing setting against published results.
>
> **Unfortunately, there is no standard benchmark in this field. Of the papers we compare to:**
>  - **Gupta et al. Use an unreleased “4K Talking Face Dataset” which we could not obtain**
>  - **StyleSync Uses VoxCeleb2 and LRW. The former has been taken down and the latter is very low quality (112x112 Pixels)**
> - **TalkLip Uses LRW and LRS2 with similar quality issues.**
> - **RadNeRF uses the dataset collected in AdNERF consisting of only one speaker.**
> - **GeneFace uses a different dataset with only 5 people from LivePortraits.**
>
> **HDTF is a higher quality and/or more diverse dataset than each of these. It is becoming a more common standard; for example, in the papers cited by the reviewer, all four use HDTF for evaluation.**
>
> The claim of applicability to "background actors" is not supported by the evaluations. The evaluation datasets, HDTF (Zhang et al., 2021) and CelebV-HQ (Zhu et al., 2022), are self-described respectively as "high-definition" and "high-quality" and feature relatively constrained pose. The claim must be removed unless it is explicitly evaluated.
>
> **We apologise for the confusion here. By background actors, we mean actors that have speaking roles for only a few lines. This used to be referred to as extras, but this term has fallen out of favour. We still expect the videos of background actors to be high-quality, but limited with data. We are more explicit in the revised version (Second Paragraph in the Introduction).**

---

> > ### Author Response · Authors · 2025-01-16
> > **Reply to Reviewer Msom (continuted)**
> >
> > The statement that "Existing models require several minutes of [subject-specific] training data." is misleading, as most visual-dubbing methods are conditioned on a single frame.
> >
> > **We have specified person-specific models here (Last contribution bullet point). This is in contrast to generic models that require a single frame. Note that we have provided comparisons to both types of model.**
> >
> > The authors claim "Our model trains faster than existing person-specific models. This is due to the deferred neural rendering priors," but it must be clarified that the comparison is in reference to NeRF-based methods, so it is not appropriate to reduce methodological differences to use of "neural rendering priors."
> >
> > **This claim is in regards to the baseline, based on the work of Thies et al. We have updated the paper to reflect this (First paragraph section 5.4).**
> >
> > The claim "We have also shown that our person-generic prior network training and adaptation strategy trains faster, reaches higher quality and works on less data than a similar model trained without priors" is misleading as it refers to an ablation study rather than comparisons with a published method.
> >
> > **We have replaced “than a similar model trained without priors” with “than an identical baseline model trained without priors” (Last paragraph Section 8).**
> >
> > It has not been demonstrated that the model produces " recognisable for for any actor." The claim should be removed.
> >
> > **We agree this claim is overstated; we will modify it to: “More able to capture recognisable idiosyncrasies”, as is evidenced by the user study metric in Table 1. (Final paragraph before contributions bullet points)**
> >
> > In claiming the proposed method is "an order of magnitude cheaper," the authors must clarify that they are referring to what they call "person-specific" methods, not those conditioned on a single frame, which are presumably orders of magnitude faster than the proposed method’s two-hour subject-specific finetuning requirement.
> >
> > **We clarify this in the contributions bullet points of the revised manuscript.**
> >
> > The related work section needs clearer task-specific organization. Many cited works address distinct tasks like avatar generation or image animation rather than visual dubbing, conflicting with the authors' characterization of them. The authors should distinguish between visual-dubbing works and those that solve distinct tasks.
> >
> > **We define visual dubbing as “the process of generating lip motions of an actor in a video to synchronise with given audio.” Similarly Neural Style Preserving Visual Dubbing, a seminal paper on the field describes it as “[a process] where the visual content is adjusted to match the new audio channel”. All of the works in these sections match this description. Furthermore, all of these works include visual dubbing works for their own baselines, in particular wav2lip, suggesting they are intended for a visual dubbing context.**
> >
> > The current related-work section organization (2.1 Person Generic Models, 2.2 Person Specific Models, and 2.3 Prior Learning for Faces) is confusing and could be more logically restructured as “Methods using single identity frames” and “Methods using multiple identity frames.” Section 2.3 appears redundant as most visual-dubbing methods in 2.2 (such as Kim et al. (2019); Song et al. (2020); and Thies et al. (2020)) can also be described as having "adopted a similar strategy of pretraining a prior model and fine-tuning to new identities."
> >
> > **This new split seems sensible, and we can change sections 2.1 and 2.2 if required. However, as stated in the first response (Weakness.1), these works do not train a prior model for appearance. Only a small number of works train a prior and we wish to highlight these in comparison to our work.**

---

> > > ### Author Response · Authors · 2025-01-16
> > > **Reply to Reviewer Msom (Continued)**
> > >
> > > Of related work, the authors claim "They all produce high-quality output but come with significant data requirements ranging from about 15 seconds Shen et al. (2022) to upwards of 10 minutes Du et al. (2023). In contrast, our method achieves similar quality using as little as 4 seconds of training data, thanks to our person-generic prior network training and person-specific adaptation." However, neither of the works referenced are compared against, nor are they visual-dubbing methods. The author should not make efficiency claims in relation to works intended to solve distinct tasks.
> > >
> > > **We have updated this to reflect the data requirements of our selected baselines (Final paragraph Section 2.2) for which we have provided comparisons.**
> > >
> > > The authors' characterization of methods conditioned on a single image as lacking in quality by nature of them being "zero-shot" ("However, existing models are either zero-shot and, therefore, lack quality") was not demonstrated and should be removed.
> > >
> > > **We have updated this to say: “However, existing person-specific models see only one frame of the actor and, therefore, lack the ability to capture identity in the form of characteristic motion and related idiosyncracies” (Abstract)**
> > >
> > > The authors' claim that subject-specific methods require "off-putting user enrollment" is not supported by citations. Which related methods require such?
> > >
> > > **This is the requirement of a few minutes of data. We will adjust this to say: “off-putting large data requirements” (Abstract)**
> > >
> > > Gupta et al. (2023) is not named Wav2LipHQ, which seems to refer only to an unrelated GitHub repository [5/A]. The authors should verify they are comparing against the intended method.
> > >
> > > **Our mistake. Wav2LipHQ is a colloquially used name, not the official name. We are comparing against Gupta et al. as we contacted the authors directly for results. We have updated all references to this in the revised version.**
> > >
> > > CelebV-HQ (Zhu et al., 2022) does not appear to be annotated as a speaking dataset. Would the authors please clarify how they selected clips for evaluation?
> > >
> > > **CelebV-HQ is annotated with actions. 77% of these are talking. We randomly selected 20 video (to match our HDTF evaluation) of those that are labelled as talking. We include this at the end of the first paragraph of section 5.**
> > >
> > > The authors write their method "can generate frames at around 5fps". Does this metric include application of the MICA (Zielonka et al., 2022) face tracker which is used during preprocessing? If accounted for, it seems the framerate should be closer to 0.2FPS. Would the authors please clarify what is included in the calculation?
> > >
> > > **This does not include MICA. The idea here being that this processing step is performed only once and inference can then be run repeatedly as 5fps. We have now explained this in the paper (Section 5.5).**
> > >
> > > The authors introduce a number of changes over previous work, such as 1) "To improve the quality of the generations, we replace the UNET used in previous works with a modified version of the StyleGAN2"; 2) "To improve temporal stability, we provide the generator with access to a window of frames surrounding the target and predict the same window of the final video"; or 3) "To encourage the network to produce better results in the lower face and mouth region, we compute masks for these areas and weigh them higher," but do not perform ablation studies. The effects of these changes should be explicitly evaluated.
> > >
> > > **The use of these components has been demonstrated in previous studies. 1) is shown in StyleSync, 2) in wav2lip and 3) in several papers such as FlashAvatar. We have included references to each in the revised paper.**
> > >
> > > To establish a fair comparison with person-generic methods, the authors should additionally evaluate with single-frame conditioning.
> > >
> > > **Our method requires more than a single frame, this is why we specified a lower limit of four seconds. For a more fair comparison with person-generic methods in our limited data setting please see EXPERIMENT 2.**
> > >
> > > Form-related points
> > >
> > > **We have addressed the form related concerns in the revised manuscript.**
> > >
> > > The authors write they "computed statistical significance in table 2," but they do not appear to have included the results.
> > >
> > > **We have included 95% Confidence intervals in Table 2. The results are significant where the lower bound of our method is greater than 50%, e.g. there is less than a 5% chance the preference is due to chance.**
> > >
> > > The nature of the work entails serious broader-impacts concerns…
> > >
> > > **The model is not able to reconstruct a person’s appearance without the corresponding neural texture. It is true that audio can be explicit; we meant that our work is not a version of face-swapping for, e.g. non-consensual pornography. We have removed this part of the claim in our discussion to reflect this.**

---

> > > > ### Comment · Reviewer_Msom · 2025-01-17
> > > >
> > > > The authors' rebuttal is appreciated.
> > > >
> > > > 1. W.1: The authors claim:
> > > >    > The critical insight of our work is the realisation that existing high-quality Neural Rendering models can benefit from a large-scale pre-training of some of their components. We call this a prior network. Our prior network is trained across multiple actors and can generalise across identities.
> > > >
> > > >    Can not the statement be considered equally true of Thies et al. (2020), or any "person-generic" visual-dubbing method? If the authors are referring to only a particular component they must limit the scope of their claims throughout, which they do not appear to have done in the revision as indicated in the rebuttal.
> > > >    If the authors decide to narrow the claims to pertain only to "person-specific" models, then they are asked about StyleSync (Guan et al., 2023):
> > > >    > The most similar work to ours could be considered to be StyleSync (Guan et al., 2023). This method performs visual dubbing and has demonstrated an ability to adapt to new identities using fine-tuning. However, the model does not decouple person-specific and person-generic components, while ours does.
> > > >
> > > >    Should the "critical insight" be restricted to "[decoupling] person-specific and person-generic components"? The authors must clarify precisely what they claim as a unique contribution.
> > > > 2. W.2: The authors must make explicit which data and in what quantities each baseline method was trained on for *each* evaluation. The reported values for StyleSync should not be expected to be identical for both the 100- and 1000-frame HDTF evaluations.
> > > > 3. W.3: Comparison against a common baseline does not make GeneFace (Ye et al., 2023) or RAD-NeRF (Tang et al., 2022) visual-dubbing methods. It is neither a fair nor interesting comparison to evaluate NeRF avatar-generation works meant to be trained on "6,000 frames" or a "5 minute video" on 100--1000-frame sequences against a method provided both ground-truth head pose and testing images with only the jaw masked. The claim that the proposed method "generalizes to limited data better than existing person-specific models" remains unsupported because no such models were included in the evaluation.
> > > > 4. W.4: If the authors are not able to find code online for the baselines, they are expected to reach out to the corresponding authors. Lack of online code does not allow the authors to claim they "compare to the most up-to-date state-of-the-art in the field."
> > > > 5. W.5: Since, of visual-dubbing works published in 2024, "all four use HDTF for evaluation," it is of reinforced importance that the authors evaluate "in an existing setting against published results."
> > > > 6. C.3: If the "claim is in regards to the baseline," the authors should state so directly. It is not correct to claim, as now written, that the authors' "model trains faster than existing person-specific models." The authors must also clarify in the text they are referring to the model "based on the work of Thies et al. [(2020)]" and not to the model itself, which differs in a number of ways. The authors must also clarify that the following statement applies only to the reproduction: "We find an order-of-magnitude speedup compared to existing person-specific models (Section 5.4), leading to a similar order-of-magnitude cost reduction."
> > > > 7. C.6: The scope of the abstract claim "at a much lower cost" must be similarly updated.

---

> > > > > ### Comment · Reviewer_Msom · 2025-01-17
> > > > >
> > > > > 8. RW.1: Both the authors' employed definition of visual dubbing and that cited appear noncontroversial. However, the authors' extrapolation of "the process of generating lip motions of an actor in a video to synchronise with given audio" to mean *any method that involves such* is not aligned with literature. Visual dubbing is an inpainting task, as exemplified by StyleSync (Guan et al., 2023), Thies et al. (2020),  Gupta et al. (2023), TalkLip (Wang et al., 2023a), Wav2Lip (Prajwal et al., 2020), Kim et al. (2019), [1], [2], [3], [4], and the authors' work, all of which are conditioned on masked frames of the test subject. It is not otherwise-unconstrained audio-conditioned avatar generation as in GeneFace (Ye et al., 2023) or RAD-NeRF (Tang et al., 2022), neither of which are presented as "visual-dubbing" methods. Can the authors point to authoritative published works presented as primarily visual-dubbing methods that do not receive non-audio information from the target frames? It does not appear realistic that an audio-only model such as GeneFace (Ye et al., 2023) would be applicable to the authors' motivating task of dubbing movie content.
> > > > > 9. RW.2: The current categorization split of person-generic and person-specific methods appears somewhat arbitrary. Take StyleSync (Guan et al., 2023) which the authors describe as "person-generic." Contrastingly, the authors of StyleSync stated of it:
> > > > >    > We take inspiration from the recent studies of inverting StyleGAN priors [1, 2, 36, 47] and propose a Personalized Optimization scheme. As audio dubbing is normally performed on speaking videos, our model can make use of only a few seconds of the person’s information and optimize additional person-specific parameters including the W+ and the generator.
> > > > >
> > > > >    The split of related work into these categories is consequential as the authors reference them when making performance claims ("generalizes to limited data better than existing person-specific models"; "order-of-magnitude speedup compared to existing person-specific models"; "significant reduction in data requirements compared to existing person-specific models"). Since it is not correct that StyleSync uses "only a single reference frame to encode the identity," it can not be considered to be purely person-generic, and if the authors do not restructure the categorization, these performance claims must be removed.
> > > > > 10. E.3: The authors are asked to report both the complete inference framerate of their method and those of the baselines. If preprocessing is necessary to apply the method, it must be included in the time calculation.
> > > > > 11. E.5: The authors' statement that their "method requires more than a single frame" is confusing as they report performance of their model trained on a single frame in Table 5 and in the caption claim "Our model is robust even on very small datasets." The authors are requested to extend the single-frame evaluation to include the baseline methods to establish a fair comparison.
> > > > > 12. F.TF.3: The text still reads that the authors "computed statistical significance in table 2" and should be modified if the authors do not report the quantity.
> > > > > 13. BI: It may not be possible to completely "reconstruct a person's appearance without the corresponding neural texture," but the authors should not claim that "Simply deleting the texture will ensure that the individual is no longer represented in the model" as the individual's data will remain in the "multi-person prior network."
> > > > > 14. The authors stated they had "addressed the form related concerns in the revised manuscript," but it should be noted that they have not addressed BC.2, BC.4, or FT.2.
> > > > >
> > > > > [1] Lee, Dongyeun, et al. "RADIO: Reference-Agnostic Dubbing Video Synthesis." Proceedings of the IEEE/CVF Winter Conference on Applications of Computer Vision. 2024.
> > > > >
> > > > > [2] Liu, Tao, et al. "DiffDub: Person-Generic Visual Dubbing Using Inpainting Renderer with Diffusion Auto-Encoder." ICASSP 2024 IEEE International Conference on Acoustics, Speech and Signal Processing (ICASSP). IEEE, 2024.
> > > > >
> > > > > [3] Liu, Kangwei, Xiaowei Yi, and Xianfeng Zhao. "ProDub: Progressive Growing of Facial Dubbing Networks for Enhanced Lip Sync and Fidelity." 2024 IEEE International Conference on Multimedia and Expo (ICME). IEEE, 2024.
> > > > >
> > > > > [4] Zhang, Longhao, et al. "PersonaTalk: Bring Attention to Your Persona in Visual Dubbing." SIGGRAPH Asia 2024 Conference Papers. 2024.

---

> > > > > > ### Author Response · Authors · 2025-01-30
> > > > > >
> > > > > > Visual dubbing is an inpainting task [...] It is not otherwise unconstrained audio-conditioned avatar generation [...], It does not appear realistic that an audio-only model such as GeneFace (Ye et al., 2023) would be applicable to the authors' task of dubbing movie content.
> > > > > >
> > > > > > **We remove these two models as comparisons, as discussed above. We clarify that person-generic models work without additional training, whereas person-specific models require at least some person-specific data of a given subject for training and/or finetuning. To fill the category of person-specific models, we include the finetuned versions of DiffDub and TalkLip (DiffDub-FT, TalkLip-FT),  as well as our baseline model without priors. Each of these models is trained on additional data and, therefore, meets our redefined classification of person-specific models.**
> > > > > >
> > > > > > The current categorization split of person-generic and person-specific methods appears somewhat arbitrary. Take StyleSync (Guan et al., 2023) which the authors describe as "person-generic." Contrastingly, the authors of StyleSync state:
> > > > > > We take inspiration from the recent studies of inverting StyleGAN priors and propose a Personalized Optimization scheme. As audio dubbing is normally performed on speaking videos, our model can make use of only a few seconds of the person’s information[...].
> > > > > >
> > > > > > **We clarify that person-generic models work without additional training, whereas person-specific models require at least some person-specific data of a given subject for training and/or finetuning. We also clarify the difference between StyleSync-G and StyleSync-P as defined in their paper in Section C.4. StyleSync-G is a person-generic model, as it requires no subject-specific training data. We also removed the Prior Learning for Faces section, as the works do not fall into the scope of the revised paper now that we look only at visual dubbing as an inpainting task.**
> > > > > >
> > > > > > The split of related work into these categories is consequential as the authors reference them when making performance claims [...] Since it is not correct that StyleSync uses "only a single reference frame to encode the identity," it can not be considered to be purely person-generic, and if the authors do not restructure the categorization, these performance claims must be removed.
> > > > > >
> > > > > > **We have modified the claim “generalizes to limited data better than existing person-specific models" to "is able to adapt to adapted to small datasets better than baselines", evidenced in Table 6, and "order-of-magnitude speedup compared to existing person-specific models" to "order-of-magnitude speedup compared to an identical baseline model trained without priors.". We also clarify we are comparing to the available StyleSync-G model**
> > > > > >
> > > > > > The authors' statement that their "method requires more than a single frame" is confusing as they report performance of their model trained on a single frame in Table 5 and in the caption claim "Our model is robust even on very small datasets." The authors are requested to extend the single-frame evaluation to include the baseline methods to establish a fair comparison.
> > > > > >
> > > > > > **Our model can use only one frame, but we do not claim that it performs well on datasets of only one frame, the results are not good with this amount of data as the model still overfits. We change the caption to "Our model is robust even on small datasets of as little as four seconds." We also remove the part of Table 5 with less than 100 frames to better reflect our claims.**
> > > > > >
> > > > > > The text still reads that the authors "computed statistical significance in table 2" and should be modified.
> > > > > >
> > > > > > **The 95% confidence intervals are included in Table 2 in brackets. We believe this validates the claim of "computed statistical significance in table 2", but for clarity we update this to be more explicit by saying "computed 95% confidence intervals in table 2".**
> > > > > >
> > > > > > It may not be possible to completely "reconstruct a person's appearance without the corresponding neural texture," but the authors should not claim that "Simply deleting the texture will ensure that the individual is no longer represented in the model" as the individual's data will remain in the "multi-person prior network."
> > > > > >
> > > > > > **We update this to say: "Simply deleting the texture will ensure that the individual’s appearance can no longer be reconstructed”, and remove the claim that they are not represented at all in the prior.**
> > > > > >
> > > > > > The authors stated they had "addressed the form related concerns in the revised manuscript," but it should be noted that they have not addressed BC.2, BC.4, or FT.2.
> > > > > >
> > > > > > **We have now addressed these, we have checked each arxiv paper for publication, added a citation for this motivational claim, and have reduced the paper size to 4MB, which is as low as we could get it without sacrificing quality.**

---

> > > > > > > ### Comment · Reviewer_Msom · 2025-02-02
> > > > > > >
> > > > > > > 1. W.2: If training code is not available, it is expected that authors would implement relevant baselines.
> > > > > > > 2. W.3: The authors' claims regarding "person-specific" models remain unsupported, as they have yet to evaluate against one. The finetuning of person-generic baselines is appreciated (though the authors are suggested to try to figure out why finetuning TalkLip increased its FVD by over 900%), but they do not make up for the lack of comparison against existing person-specific work. Of particular relevance is StyleSync-P (Guan et al., 2023), which, while trained over a broad multi-subject corpus, similarly "performs visual dubbing and has demonstrated an ability to adapt to new identities using fine-tuning." That the proposed method only clearly outperforms the generic StyleSync-G (Guan et al., 2023) when provided with 100,000% the target-subject data further highlights the necessity of the comparison.
> > > > > > > 3. W.4: The authors do not appear to have made the indicated adjustment, as they still claim throughout the paper to compare to the "state-of-the-art":
> > > > > > >    1. "we achieve state-of-the-art in terms"
> > > > > > >    2. "we achieve state-of-the-art in this respect"
> > > > > > >    3. "our method achieves state-of-the-art"
> > > > > > >    4. "comparisons of our model with state-of-the-art"
> > > > > > >    5. "comparisons to the state-of-the-art"
> > > > > > >    6. "5.1 Comparisons to State-of-the-Art"
> > > > > > >    7. "compare our model to the state-of-the-art"
> > > > > > >    8. "comparisons to state-of-the-art"
> > > > > > >    9. "our method achieves state-of-the-art"
> > > > > > >    10. "our model archives (sic) state-of-the-art"
> > > > > > >    11. "our model is indeed state-of-the-art"
> > > > > > >    12. "compare our method to the existing state-of-the-art"
> > > > > > >    13. "comparisons with recent state of the art"
> > > > > > > 4. W.5: The authors have not yet laid the foundation necessary to establish that CelebV-HQ might act as an "evaluation set on which all models may be compared fairly." In order to claim such, the authors should at a minimum compute similarity metrics between the datasets, such as FID, and include a brief discussion in the text on dataset sources.
> > > > > > > 5. RW.1/2: It does not appear that the authors have updated the related-work section, which continues to incorrectly qualify a number of works as visual-dubbing methods. The issue is not that the authors shouldn't cite works that target adjacent problems, but that if a method does more than inpaint the lower portion of a face -- or requires more than an audio signal to drive -- it is not directly comparable, and its characterization invites confusion.
> > > > > > > 6. E.5: It is disappointing that, rather than establish a fair comparison by extending the single-frame evaluation to include the baseline methods, the authors have decided to remove experiments from their paper. That "the results are not good" makes it all-the-more important that the full evaluation be included in order to properly contextualize the work.

---

> > > > > ### Author Response · Authors · 2025-01-30
> > > > >
> > > > > Thank you again for your detailed feedback, we feel we have been able to more accurately place our work in the context of visual dubbing and to make more accurate claims. We will again address the reviewer's comments point by point:
> > > > >
> > > > > W.1: Should the "critical insight" be restricted to "[decoupling] person-specific and person-generic components"? The authors must clarify precisely what they claim as a unique contribution.
> > > > >
> > > > > **We thank the reviewer for this suggestion regarding the claim. We clarify our contributions more accurately and change this “critical insight” to say as suggested. It now reads: “The critical insight of our work is that Neural Rendering models can be de-coupled into person-generic and person-specific components and that they benefit from a large-scale pre-training of generic ones.”**
> > > > >
> > > > > W.2: The authors must make explicit which data and in what quantities each baseline method was trained on for each evaluation. The reported values for StyleSync should not be expected to be identical for both the 100- and 1000-frame HDTF evaluations.
> > > > >
> > > > > **We add this detail in section C.4 and reference it in the main text. The authors of StyleSync would only agree to share results based on their generic model (e.g. they would not perform their finetuning or share the code with us), this is labelled Ours-G in the StyleSync paper. For this reason, we should expect the 100 and 1000 frame evaluation to be the same, as this is a generic model. We have updated references to this baseline to say StyleSync-G for clarity and have discussed exactly what this means in Section C.4.**
> > > > >
> > > > > W.3: Comparison against a common baseline does not make GeneFace (Ye et al., 2023) or RAD-NeRF (Tang et al., 2022) visual-dubbing methods. It is neither a fair nor interesting comparison to evaluate NeRF avatar-generation works meant to be trained on "6,000 frames" or a "5 minute video" on 100--1000-frame sequences against a method provided both ground-truth head pose and testing images with only the jaw masked. The claim that the proposed method "generalizes to limited data better than existing person-specific models" remains unsupported because no such models were included in the evaluation.
> > > > >
> > > > > **Following consideration, we agree with the reviewer on this point. These are not entirely fair comparisons. While we do think they address a somewhat similar task, we agree that the evaluation is not entirely fair, as these method have more area to generate. We now only consider methods that replace the lower part of the face. We remove the RaDNeRF and GeneFace papers from our evaluation and change the "generalizes to limited data better than existing person-specific models" claim to  "is able to adapt to small datasets better than baselines" as evidenced in Table 1. Following this significant change, we have also updated Figures 5 & 6, Table 1 now replaces Table 6 with all the new experiments, and the video is also similarly updated.**
> > > > >
> > > > > W.4: If the authors are not able to find code online for the baselines, they are expected to reach out to the corresponding authors. Lack of online code does not allow the authors to claim they "compare to the most up-to-date state-of-the-art in the field."
> > > > >
> > > > > **We remove this claim from the paper and the video.**
> > > > >
> > > > > W.5: Since, of visual-dubbing works published in 2024, "all four use HDTF for evaluation," it is of reinforced importance that the authors evaluate "in an existing setting against published results."
> > > > >
> > > > > **We also use HDTF for evaluation in an existing setting against published results. DiffDub and MuseTalk, which we both use as baselines are evaluated in the same setting. Currently there does not exist a dataset on which all baselines have been tested. However, as no models we are comparing to have been trained on CelebV-HQ, this acts as an unseen evaluation set on which all models may be compared fairly.**
> > > > >
> > > > > C.3: If the "claim is in regards to the baseline," the authors should state so directly. It is not correct to claim, as now written, that the authors' "model trains faster than existing person-specific models." The authors must also clarify in the text they are referring to the model "based on the work of Thies et al. [(2020)]" and not to the model itself, which differs in a number of ways. The authors must also clarify that the following statement applies only to the reproduction: "We find an order-of-magnitude speedup compared to existing person-specific models (Section 5.4), leading to a similar order-of-magnitude cost reduction." The scope of the abstract claim "at a much lower cost" must be similarly updated.
> > > > >
> > > > > **We have done this in the newer revised version. Removing the cost claim in the abstract and updating the claim “We find an order-of-magnitude speedup compared to existing person-specific models (Section 5.4), leading to a similar order-of-magnitude cost reduction." To say: “We find an order-of-magnitude speedup compared to a baseline model trained without priors”**

---

> ### Author Response · Authors · 2025-02-03
>
> 1. We do not have the resources to implement many baselines where code is unavailable. This is very non-standard in this field. For example, please see the recent PAMI motion [1] that discourages reviewers from asking for comparisons to closed-source methods. We can appreciate that a comparison to StyleSync-P may be merited, but there are issues with implementing this, as detailed below. The other baselines, e.g. RADIO, ProDub and PersonaTalk, would fall under the remit of the passed motion.
>
>
> 2. Following the re-categorisation of person-generic and person-specific models, we have compared to models that show additional individual data (the -FT models). We have also compared to the baseline model. Again, StyleSync-P is not publically available with code/or data. Details are also missing from the paper, making it challenging to re-implement properly. For example, they state: "We process all videos at 25 fps and align all faces according to pre-detected landmarks at the eyes. All faces are cropped to the size of 256 × 256. A same U-shape mask is adopted as shown in Fig. 2 to erase the mouth, cheeks and jaws at the target frame." [2] It is unclear precisely what this alignment process should be or how to generate the U-shaped mask. Furthermore, there are no details on batch size, learning rate, optimiser, number of training epochs/iterations, number of GPUs and types, or random seeds essential for a proper re-implementation. Would the reviewer consider it acceptable to compare with a version of this model, trained on the same HDTF training set we use and done with our best guesses of these details? It is likely we will not obtain results of the same quality as StyleSync if the authors have tuned their hyperparameters, and we cannot.
>
>
> 3. The reviewer has stated: "Claims relating to "state-of-the-art" must be removed if the authors do not evaluate their proposed model against recent, published methods." We have performed these comparisons now and so do keep the state-of-the-art claim. We have removed the claim **most up-to-date** state of the art in the supplementary video as requested.
>
>
> 4. We have included a dataset source discussion. Computing similarity between datasets is a good idea but difficult as VoxCeleb2 has been withdrawn as a dataset, so it is not easy to see how similar the StyleSync training data is compared to CelebV-HQ. We, however, run a small experiment with HDTF and CelebV-HQ. We randomly select one frame from each video in each dataset and compare the FID scores; this gives a value of 350. When we compare HDTF to itself similarly, the value is 87, and Celeb-V to itself is 40, suggesting there is more difference intra-dataset than inter-dataset. We have requested access to LRW and will repeat this experiment with this dataset if it is granted.
>
>
> 5. We have updated this section to include only works meeting the reviewer's criteria.
>
>
> 6. To better contextualise this limitation, we discuss how our work does not perform well with smaller amounts of data in the limitations section.
>
>
> [1] https://tc.computer.org/tcpami/tc-motions/
>
> [2] StyleSync: High-Fidelity Generalized and Personalized Lip Sync in Style-based Generator, Guan et al. CVPR23

---

### Review · Reviewer_fw5J · 2025-01-04

**Summary Of Contributions:**

This paper presents a method for visual dubbing that synchronizes lip movements of actors in videos with given audio. The approach first pretrains a multi-person prior network and then performs person-specific finetuning to adapt to new identities using a few seconds of data. The model achieves state-of-the-art results in visual quality, lip sync, and idiosyncracies, and is more data-efficient than prior methods.

**Audience:**

Yes

**Claims And Evidence:**

Yes

**Requested Changes:**

See weaknesses

**Strengths And Weaknesses:**

Strengths:
1. The training time for the method to adapt to a new identity is significantly lower than person-specific methods.
2. The method achieves higher visual dubbing quality compared to various baselines.


Weaknesses:
1. While the method emphasizes data efficiency, the person-specific finetuning still requires including a generic dataset and mixing it with the person-specific data. How did the author determine the 1:1 mixing ratio and how will this ratio affect the quality of the outputs?
2. When comparing against person-generic methods, is it possible to also finetune the person-generic baselines using the same data (mixture of generic and person-specific data) as the proposed method and see if there are improvements on the baseline methods? Right now the comparison seems to be a bit unfair.
3. The quantitative evaluations only considered per-frame metrics (FID, PSNR, SSIM). Video quality metrics such as FVD[1] should also be considered to assess the overall video quality.

[1] Unterthiner, Thomas, et al. "FVD: A new metric for video generation." (2019).

---

> ### Author Response · Authors · 2025-01-16
> **Reply to Reviewer fw5J**
>
> **We would like to thank the reviewer for this thoughtful and insightful feedback. We will address comments here point-by-point.**
>
> While the method emphasizes data efficiency, the person-specific finetuning still requires including a generic dataset and mixing it with the person-specific data. How did the author determine the 1:1 mixing ratio and how will this ratio affect the quality of the outputs?
>
> **The data efficiency claim applies to the amount of person-specific data required for an individual. The ratio was taken following examples in previous work, for example StyleSync uses a 1:1 ratio, as do several works in NLP, e.g. (Haque et al 2020).**
>
> ***Haque, R., Moslem, Y., & Way, A. (2020). Terminology-Aware Sentence Mining for NMT Domain Adaptation: ADAPT’s Submission to the Adap-MT 2020 English-to-Hindi AI Translation Shared Task. Proceedings of the 17th International Conference on Natural Language Processing (ICON): Adap-MT 2020 Shared Task, 17–23. https://aclanthology.org/2020.icon-adapmt.4***
>
> When comparing against person-generic methods, is it possible to also finetune the person-generic baselines using the same data (mixture of generic and person-specific data) as the proposed method and see if there are improvements on the baseline methods? Right now the comparison seems to be a bit unfair.
>
> **This is a good suggestion. We have done this as far as possible based on training code releases in EXPERIMENT 2. It can be seen that our model is able to use the finetuning data better than the person-generic baselines.**
>
> The quantitative evaluations only considered per-frame metrics (FID, PSNR, SSIM). Video quality metrics such as FVD[1] should also be considered to assess the overall video quality.
>
> **We show FVD in EXPERIMENT 3. Our method achieves state-of-the-art in this metric for all datasets. We thank the reviewer for helping make our claim of state-of-the-art stronger by suggesting this metric.**

---

> > ### Author Response · Authors · 2025-02-07
> > **Reply to Reviewer fw5J**
> >
> > Thank you again for your detailed review. We now have made several revisions to improve our paper. Have we adequately addressed your concerns and/or have you got any further points you wish to discuss?
> >
> > Thanks,
> > Paper 3824 Authors

---

### Review · Reviewer_7UhW · 2025-01-12

**Summary Of Contributions:**

This paper studies visual dubbing, a process to generate lip motions of a human in a video given an audio. It proposes a meta-learning-like method, to train a large prior network that can be applied to new people through a few-shot training. The experiments show that the proposed model achieves state-of-the-art in both visual quality and recognizability.

**Audience:**

No

**Claims And Evidence:**

Yes

**Requested Changes:**

Please see "Weaknesses". Other requested changes:

- Given that the FID decreases when adding new components, I would like to see the qualitative visualizations of ablation variants to verify the claim "The post-processing increases
FID suggests worse visual quality. However, the user ratings show a preference for post-processing."
    - The supplementary video provides these results, but it is better to (1) put the visualizations in the main paper, and (2) show a zoomed image of the specific differences to strengthen this claim.
    - Also, please consider the variant "Ours without both methods", to see how the FID will be.
- Question: The idea to train a prior model to support few-shot training for actual use is in a similar way to meta-learning. What is the relationship between this proposed method and meta-learning, what are common and what are different? Will it be helpful to apply meta-learning methods in (meta-)training?

**Strengths And Weaknesses:**

### Strengths
- The proposed model consistently outperforms all the baselines in various benchmarks. Also, various quantitative measurements are introduced to evaluate the model in a holistic manner.
- From the qualitative results, the proposed method generates reasonable lip motion that is very close to the real data with few artifacts.

### Weaknesses
- All the baselines are published at least one year ago (before or in 2023). I wonder if this means that there was no follow-up work in 2024, so there will be few "audiences" interested in this topic; or if the authors are not extensively considering the latest publications, which weaken the "evidence" of this paper.
- Table 1 is messy, with inconsistent formats
    - In most columns, the second best is marked as yellow,. However, in three specific columns, HDTF-SSIM, CelebV-SSIM, and CelebV-Lip, there are no yellow-marked numbers. Instead, there is an underlined number.
    - The "Lip" and "ID" from HDTF are marked as blue and red. I did not find any follow-up explanations.
- I wonder what is the actual input and output of the proposed model. Where does the 3D prior come from, from the dataset or from some depth estimation methods? And how to align this with the actual usage in the evaluation?
- The motivation for the model's design is not clear. In most of the methods part, the authors only introduced the method without providing the insight behind of the reason.
    - The proposed method uses a StyleGAN generator. Given that the pre-trained diffusion model (1) is used by baselines, (2) has sufficiently strong prior to understand various conditions and generate photorealistic images, and (3) also has various way to speed up for high efficiency, I would like to know the reason to choose StyleGAN instead of diffusion models. It seems that StyleGAN introduces the limitation of resolution of the output results.
- I wonder how the proposed method ensures the video smoothness of the output video.
- I wonder whether the proposed method supports various face conditions, e.g., faces and lips with scars, face painting, or accessories.
- The pipeline figures (Figs. 3,4) are blurred with aliased texts - they seem to be screenshots at low resolution. Also, I wonder if the proposed method supports various face poses or motions (e.g., talking when lying down on their side, or talking along with the getting-up process), other than a simple, static pose with minor motions.
- (Minor) Incorrect spelling cases: UNET should be UNet, NERF should be NeRF.

---

> ### Author Response · Authors · 2025-01-16
> **Reply to Reviewer 7UhW**
>
> **We would like to thank the reviewer for this thoughtful and insightful feedback. We will address comments here point-by-point.**
>
> All the baselines are published at least one year ago (before or in 2023). I wonder if this means that there was no follow-up work in 2024, so there will be few "audiences" interested in this topic; or if the authors are not extensively considering the latest publications, which weaken the "evidence" of this paper.
>
> **Please see EXPERIMENT 1, which includes some more recent results, including two models from 2024. We believe this better positions our paper against the recent state of the art and should help improve the audience and evidence of this work**
>
> Table 1 is messy, with inconsistent formats[…]
>
> **This was a mistake on our part. We have now fixed this Table, including consistent best/second best formatting and have removed this unused colour coding.**
>
> What is the actual input and output of the proposed model. Where does the 3D prior come from?[...]
>
> **The input and output are shown in Figures 3 & 4. The 3D data comes from monocular reconstruction; we describe this process in more detail in Appendix A.1**
>
> The motivation for the model's design is not clear[...] The proposed method uses a StyleGAN generator. Given that the pre-trained diffusion model (1) is used by baselines, (2) has sufficiently strong prior to understand various conditions and generate photorealistic images, and (3) also has various way to speed up for high efficiency, I would like to know the reason to choose StyleGAN instead of diffusion models.
>
> **We chose a StyleGAN following the results demonstrated in StyleSync (Gaun et al. 2023). The idea of using diffusion is a good one and one that we want to pursue in future work. At the time of starting this project, we found diffusion models too slow to be cost-effective. The recent usage of pre trained LDMs (Such as EMO (Tian et al. 2024) would solve this, and we are investigating it.**
>
> I wonder how the proposed method ensures the video smoothness of the output video.
>
> **We pass the model several frames in a temporal window, generate over this same window and have a discriminator look at the whole window. This produces good stability as shown in Wav2Lip (Prajwal et al. 2020).
> I wonder whether the proposed method supports various face conditions, e.g., faces and lips with scars, face painting, or accessories.
> Our model should be able to handle this fairly well. The neural texture has the ability to reconstruct these directly through its optimization. Although face paint that is not seen in the prior may cause some issues.**
>
> The pipeline figures (Figs. 3,4) are blurred with aliased texts - they seem to be screenshots at low resolution.
>
> **We have converted these Figures to vector graphics which should appear better.**
>
> Also, I wonder if the proposed method supports various face poses or motions (e.g., talking when lying down on their side, or talking along with the getting-up process), other than a simple, static pose with minor motions.
>
> **The CelebV-HQ results show better pose diversity. We show some of this in the supplementary Figure 10 Section C.3. We find that our model handles up to around 30-degree variations in pose well, despite our prior being trained on most frontal facing video. We have not tested on faces lying down or on their side; it is possible that the face tracking may fail at this point and/or the input may be too different from the prior. However, the faces could be aligned in 2D in these cases, and our model should still work.**
>
> (Minor) Incorrect spelling cases: UNET should be UNet, NERF should be NeRF.
>
> **We have fixed these in the revised version.**
>
> I would like to see the qualitative visualizations of ablation variants to verify the claim "The post-processing increases FID suggests worse visual quality. However, the user ratings show a preference for post-processing." The supplementary video provides these results, but it is better to (1) put the visualizations in the main paper, and (2) show a zoomed image of the specific differences to strengthen this claim.
>
> **We have included such a figure in the supplementary in Figure 9, section C.2. This shows how artefacts are reduced in the border region between the face and background**
>
> Question: The idea to train a prior model to support few-shot training for actual use is in a similar way to meta-learning. What is the relationship between this proposed method and meta-learning, what are common and what are different? Will it be helpful to apply meta-learning methods in (meta-)training?
>
> **This is true; including methods from the meta-learning literature are of great interest, for example, LoRA. This is an emphasis of our future work.**

---

> > ### Author Response · Authors · 2025-02-07
> > **Reply to Reviewer 7UhW**
> >
> > Thank you again for your detailed review. We now have made several revisions to improve our paper. Have we adequately addressed your concerns and/or have you got any further points you wish to discuss?
> >
> > Thanks,
> > Paper 3824 Authors

---

### Author Response · Authors · 2025-01-16
**Reply to All Reviewers**

First, we would like to thank you all for this highly detailed feedback. We are glad to hear that reviewers find our work looks visually appealing (All Reviewers), outperforms baselines (7UhW, fw5J), trains faster (fw5J), and that our paper is well-written (Msom). We will post individual, point-by-point replies to each reviewer. To avoid repetition, however, we include the additional experiments we run here. We label these as EXPERIMENTS 1, 2 and 3 and will refer to them in individual replies. For clarity, in the revised manuscript, changes are highlighted in bold red text.

EXPERIMENT 1: More Recent Baselines. To compare models released in 2024, we also included results from DiffDub and MuseTalk. We have added them to Appendix C (Table 6) in the revised paper. We include the quantitative metrics only due to time constraints. If accepted, we will re-run the user study to get qualitative metrics and add all the new baselines to Table 1 in the Camera-ready version.

EXPERIMENT 2: Finetuning Generic Baselines. Of the person-generic baselines, we are unable to finetune StyleSync or Gupta et al. as the authors provided the results, and they will not commit the compute to do the finetuning. Furthermore, MuseTalk did not release training code. For TalkLip and the now included DiffDub, we show the results of finetuned models in Table 6, Appendix C. These have the suffix -FT for finetuned and are now listed in the person-specific models as they have been trained on personal data. We perform finetuning using the same losses, learning rates and batch sizes used in the training code for these papers.

EXPERIMENT 3: FVD, we include FVD for all evaluations. This metric is shown in Tables 1, 4 and 6. Our model achieves state-of-the-art in this metric.

---

> ### Comment · Reviewer_Msom · 2025-01-16
>
> In their response, the authors appear to have confirmed the baselines were not trained on the same data as the proposed method. It is, consequently, difficult to evaluate any performance or efficiency claims.

---

> ### Author Response · Authors · 2025-01-30
>
> Thank you for your quick response. The baselines DiffDub and MuseTalk have been trained using the same training split of HDTF and so can be compared directly for our claims. Models labelled -FT, and the person-specific NeRF papers in the revised version have also been finetuned on the same person-specific data as our model.
>
> It is not possible to compare other methods using the same training data, as none of these provide training code. For StyleSync and Gupta et al., we had to directly ask for evaluation results using pretrained models, which they were happy to provide in comparison to our method. Still, we are not able to retrain them, as they do not wish to share code to avoid competition.
>
> It is worth noting that it is not standard in this field to require all baselines to be trained on the same data, as for ethical and commercial reasons, the vast majority of methods are either closed source or provide inference code and a pre-trained model only, without training code. Many of the accepted papers the reviewer has cited have also compared with methods without training code, using pre-trained models trained on different datasets; DiffDub compares with DAETalker [1] (Closed Source), ProDub with SyncTalkFace [2] (Closed Source), PersonaTalk with VideoReTalking [3] (No Training Code, only pre-trained model). This is also true of recent CVPR papers not mentioned, for example, Learning Dynamic Tetrahedra [4] (Zhang et al) compare to Audio2Head [5] (Pretrained only, no training code). Many methods compare to SadTalker [6], which is also inference only (e.g. Faces that Speak [7]; EDTalk [8]), without evaluating on the same test set.
>
> It would be infeasible to compare to a sufficient number of recent baselines if reimplementing and retraining each on the same dataset was required.
>
> Furthermore, the evaluation on CelebV-HQ, on which none of the models, ours or baseline, have trained on, acts as an unbiased evaluation set for all models.
>
> [1] DAE-Talker: High Fidelity Speech-Driven Talking Face Generation with Diffusion Autoencoder, Du et al. ACM Multimedia 2023
>
> [2] SyncTalkFace: Talking Face Generation with Precise Lip-Syncing via Audio-Lip Memory, Park et al. AAAI 2023
>
> [3] VideoReTalking: Audio-based Lip Synchronization for Talking Head Video Editing in the Wild, Cheng et al. SIGGRAPH Asia 2023
>
> [4] Learning Dynamic Tetrahedra for High-Quality Talking Head Synthesis, Zhang et al. CVPR 2024
>
> [5] Audio2Head: Audio-driven One-shot Talking-head Generation with Natural Head Motion (IJCAI 2021), Wang et al. IJCAI 2021
>
> [6] SadTalker: Learning Realistic 3D Motion Coefficients for Stylized Audio-Driven Single Image Talking Face Animation, Zhang et al. CVPR 2023
>
> [7] Faces that Speak: Jointly Synthesising Talking Face and Speech from Text, Jang et al. CVPR 2024
>
> [8] EDTalk: Efficient Disentanglement for Emotional Talking Head Synthesis, Tan et al. ECCV 2024

---

> > ### Comment · Reviewer_Msom · 2025-02-02
> >
> > Regardless of what is permitted at other venues, it remains difficult to evaluate the authors' performance and efficiency claims when the baselines are not trained on the same data.
> >
> > StyleSync-G (Guan et al., 2023) outperforms the proposed method in two out of seven metrics and ties in two others when provided only 1% the amount of target-subject data (1 vs. 100 frames). Can the authors state with any confidence that the improvements observed in the other three metrics can be attributed to anything other than differences in data?

---

> > > ### Author Response · Authors · 2025-02-02
> > >
> > > In addition to the HDTF results, we also provide results on CelebV-HQ, where we outperform StyleSync-G on all metrics (Table 1) and a user study which directly compares our model to StyleSync-G in a forced choice experiment, in which we outperform this method with >95% confidence, as shown that the lower bound of the 95% CI for Ours>StyleSync-G is greater than 50% (Table 2). These additional comparisons (CelebV-HQ and the User Study) are performed with datasets neither model has been trained on. We also show the results qualitatively in Figure 1 and the supplementary video. When combining all of these, we are confident that we have sufficient evidence that our model does outperform StyleSync-G.

---

### Decision · Action_Editor_QR9Z · 2025-02-22

**Recommendation:** Reject

**Comment:**

## Main point of discussion

To understand the arguments of the author and the reviewers. Some context is provided.

The authors categorize existing work based on the amount of visual training data into two groups:

- Person-generic: The subject of interest is not present in the training data. Only a single frame of the subject is provided during inference.
- Person-specific: The model is trained on abundant data of the subject of interest (e.g., 2-5 minutes of footage).

The proposed method involves learning a prior (similar to a pretrained foundation model), which can be fine-tuned to dub a specific person. As a result, the method can handle both person-generic and person-specific dubbing.

In terms of comparison, the authors primarily focus on person-generic methods but use 100-1000 frames of the person of interest, as opposed to just a single frame in person-generic method. In the rebuttal, the authors provide results under the one-frame condition but later remove them due to poor visual quality.

During the discussion, the reviewers express differing opinions regarding the sufficiency of the comparisons. While one reviewer finds the comparison approach acceptable, Reviewer Msom raises concerns, believing that such a comparison is necessary since the current performance gain over the single-frame baseline is insignificant.
Furthermore, the reviewers also question the comparison to the person-specific baseline. They suggests that, in addition to the current fine-tuning approach, the authors should fine-tune two person-generic methods to provide a more comprehensive comparison.



*It is worth noting that the AE does not place significant weight on the discussion surrounding the state-of-the-art quality claim, as it requires comparisons with closed-source baselines. The AE acknowledges the various challenges associated with making such comparisons.*



## Final Thoughts

In the end, two reviewers lean toward accepting the manuscript, while one reviewer strongly opposes it. Although the reviewers praise the generation quality, they all agree that the novelty of the approach is limited.

The outstanding concerns are central to the main claim of verifying the method’s effectiveness.
Considering the arguments from both sides, the AE cannot confidently conclude that these issues are fully resolved. For instance, evaluating the method under the single-frame condition seems necessary, as it may be commonly used in practice. Modern video foundation models can already be fine-tuned to generate a clip of person from a single image (e.g., image-to-video). Therefore, the single-frame results should be included in the submission to provide readers with a more comprehensive understanding.

In view of the above, while this manuscript shows potential, it may be premature to accept it for publication in TMLR at its current stage.

**Audience:**

This paper’s technique is directly beneficial to a small community working on visual dubbing.
The topic may interest a subfield of the audience focused on audio-video generation, but only to some extent.

**Claims And Evidence:**

This paper addresses the visual dubbing problem, where, given an audio input, the method generates a video synchronized to the audio focusing on the generation of lip and mouth movements .

The authors propose a method that involves learning a prior (similar to a pretrained foundation model), which can be fine-tuned to dub a specific person while reducing the amount of data required for that individual.


## Claim and evidence

The novelty of this work, compared to existing approaches, lies in a specially designed decoupling method. The main claims of this paper, after revision, are:

1. A post-processing algorithm
1. A visual dubbing model capable of producing high-quality, idiosyncratic results while enabling data-efficient fine-tuning for dubbing
1. State-of-the-art generation quality with improved efficiency compared to person-specific models


The paper appears to provide substantial evidence supporting the first claim. Following the revision, extensive discussion has focused on the effectiveness of the method, in particular the sufficiency of the provided empirical verification, and whether the state-of-the-art performance is adequately substantiated.

**Resubmission Of Major Revision:**

The authors may consider submitting a major revision at a later time.